# No "Zero-Shot" Without Exponential Data: Pretraining Concept Frequency Determines Multimodal Model Performance

**Vishaal Udandarao**[*1,2]    **Ameya Prabhu**[*1,3]    **Adhiraj Ghosh**[1]    **Yash Sharma**[1]
**Philip H.S. Torr**[3]    **Adel Bibi**[3]    **Samuel Albanie**[2†]    **Matthias Bethge**[1†]
[1]Tübingen AI Center, University of Tübingen  [2]University of Cambridge  [3]University of Oxford

 github.com/bethgelab/frequency_determines_performance
 huggingface.co/datasets/bethgelab/let-it-wag

## Abstract

Web-crawled datasets underlie the impressive "zero-shot" performance of multimodal models, such as CLIP for classification and Stable-Diffusion for image generation. However, it is unclear how meaningful the notion of "zero-shot" *generalization* is for such models because the extent to which their pretraining datasets encompass downstream concepts used in "zero-shot" evaluation is unknown. In this work, we ask: *How is the performance of multimodal models on downstream concepts influenced by the frequency of these concepts in their pretraining datasets?*

We comprehensively investigate this question across 34 models and 5 standard pretraining datasets, generating over 300GB of data artifacts. We consistently find that, far from exhibiting "zero-shot" generalization, multimodal models require exponentially more data to achieve linear improvements in downstream "zero-shot" performance, following a sample inefficient log-linear scaling trend. This trend persists even when controlling for sample-level similarity between pretraining and evaluation datasets [81], and testing on purely synthetic data distributions [52]. Furthermore, upon benchmarking models on long-tailed data sampled based on our analysis, we demonstrate that multimodal models across the board perform poorly. We contribute this long-tail test dataset as the *Let it Wag!* benchmark to further research in this direction. Taken together, our study reveals an exponential need for training data which implies that the key to "zero-shot" generalization capabilities under large-scale training data and compute paradigms remains to be found.

## 1    Introduction

Multimodal models like CLIP [98] and Stable Diffusion [104] have revolutionized performance on downstream tasks. CLIP is now the *de facto* standard for "zero-shot" image recognition [143, 74, 136, 49, 142] and image-text retrieval [47, 64, 25, 127, 139], while Stable Diffusion is now the *de facto* standard for "zero-shot" text-to-image (T2I) generation [100, 18, 104, 42]. In this work, we investigate this empirical success through the lens of zero-shot generalization [70], which refers to the ability of models to apply their learned knowledge to new unseen concepts (not seen during training). Accordingly, we ask: *Are current multimodal models truly capable of "zero-shot" generalization?*

To tackle this question, we conduct a comparative analysis involving two main factors: (1) the performance of models across various downstream tasks, and (2) the frequency of test concepts within their pretraining datasets. We compile a comprehensive list of $4,029$ concepts[2] from 27 downstream

---

[*]equal contribution, † equal supervising

[2]class categories for classification tasks, objects in the text captions for retrieval tasks, and objects in the text prompts for generation tasks, see Sec. 2 for more details on how we define concepts.

38th Conference on Neural Information Processing Systems (NeurIPS 2024).

tasks spanning classification, retrieval, and image generation, assessing model performance against these concepts. Our analysis spanned five large-scale image-text pretraining datasets with different scales, data curation methods and sources (CC-3M [115], CC-12M [28], YFCC-15M [123], LAION-Aesthetics [111], LAION-400M [110]), and evaluated the performance of 10 CLIP models and 24 T2I models, spanning different architectures and parameter scales. We consistently find across our experiments that, across concepts, the frequency of a concept in the pretraining dataset is *a strong predictor* of the model's performance on test examples containing that concept (see Fig. 2). Notably, ***model performance scales linearly as the concept frequency in pretraining data grows exponentially*** *i.e.*, we observe a consistent log-linear scaling trend. We find that this log-linear trend is robust to controlling for correlated factors (similar samples in pretraining and test data [81]) and testing across different concept distributions along with samples generated entirely synthetically [52].

Our findings indicate that the impressive empirical performance of multimodal models like CLIP and Stable Diffusion can be largely attributed to the presence of test concepts within their vast pretraining datasets, thus their reported empirical performance does not constitute "zero-shot" generalization. Quite the contrary, these models require exponentially more data on a concept to linearly improve their performance on tasks pertaining to that concept, suggesting significant sample inefficiency.

We additionally document the distribution of concepts encountered in pretraining data and find that:

- **Concept Distribution:** Across all pretraining datasets, the distribution of concepts is long-tailed (see Fig. 5), indicating that a large fraction of concepts are rare. Given the extreme sample inefficiency observed, these rare concepts are not properly learned during pretraining.

- **Concept Correlation across Pretraining Datasets:** The distributions of concepts across different pretraining datasets are strongly correlated (see Tab. 4), suggesting that web crawls yield surprisingly similar concept distributions across very diverse data curation strategies. This necessitates explicit concept rebalancing efforts explored in prior work [11, 135].

- **Image-Text Misalignment in Pretraining Data:** Concepts often appear in one modality but not the other, implying significant misalignment (see Tab. 3). Our released data artifacts can help image-text alignment efforts at scale by precisely indicating examples where modalities misalign. Note that the log-linear trend across both modalities is robust to this misalignment.

To provide a simple benchmark for multimodal generalization that controls for concept frequency in the pretraining set, we introduce a new long-tailed test set, "*Let It Wag!*". Current models trained on both openly available datasets (*e.g.*, LAION-2B [111], DataComp-1B [47]) and closed-source datasets (*e.g.*, OpenAI-WIT [98], WebLI [30]) have significant drops in performance (see Fig. 6), suggesting that our findings may also transfer to closed-source datasets. We publicly release all data artifacts, amortising the cost of analyzing image-text pretraining datasets for future efforts focused on a more data-centric understanding of the properties of multimodal models.

**Situating our Contributions in Broader Literature.** Our comprehensive analysis of several image-text datasets significantly adds to prior investigations on the role of pretraining data in affecting performance for both CLIP [92, 81, 43] and language models [62, 102, 82], by (1) showing that concept frequency determines zero-shot performance, and (2) pinpointing the exponential need for training data as a fundamental issue for current multimodal foundation models. We conclude that the key to "zero-shot" generalization under large-scale training paradigms remains to be found.

## 2 Concepts in Pretraining Data and Quantifying Frequency

In this section, we discuss how to estimate concept frequencies within pretraining datasets. We first define our concepts of interest, then describe algorithms for extracting their individual frequencies from images and text captions of pretraining datasets independently, and describe how we aggregate them to compute matched image-text concept frequencies. For a schematic overview, see Fig. 1.

**Defining Concepts.** We define "concepts" as the specific objects/relations we seek to analyze in pretraining datasets. Since our goal is to analyze downstream performance of models, we source concepts from 27 target evaluation datasets. For zero-shot classification datasets, extracted concepts are class names, such as the $1,000$ object classes in ImageNet [36] (*e.g.*, tench, goldfish). We also include relational verbs and verb-noun combinations since they are the classes of the UCF101 dataset [116] (*e.g.*, diving, brushing teeth) as well as background nouns from the SUN397 dataset [133] (*e.g.*, abbey, sky). For retrieval and image generation datasets, concepts are all nouns in test set captions or

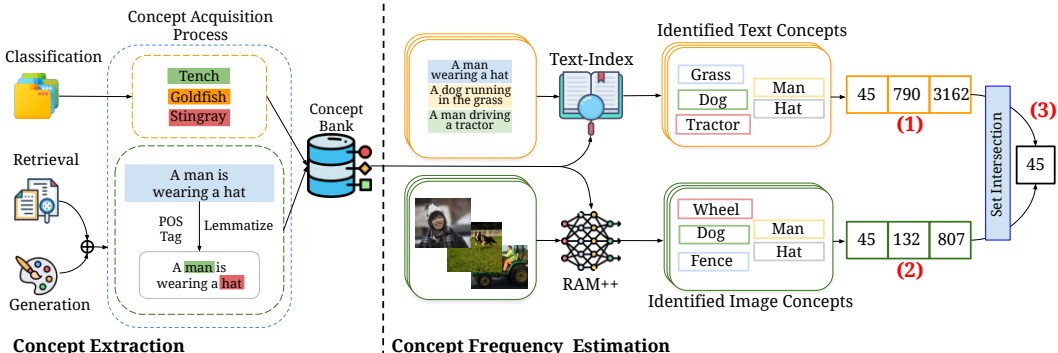

Figure 1: **Concept Extraction and Frequency Estimation.** (*left*) We compile $4,029$ concepts from 27 evaluation datasets. (*right*) We construct efficient indices for text-search (unigram indexing (1)) and image-search (RAM++ (2)); intersecting hits from both gives (3) image-text matched frequencies.

generation prompts. For example, from, "A man is wearing a hat", we extract "man" and "hat" as concepts. We filter out ambiguous or irrelevant concepts that are present in less than five downstream evaluation samples. In sum, we collate $4,029$ concepts sourced from 17 classification, 2 image-text retrieval, and 8 text-to-image generation downstream datasets (see Tab. 1 and Sec. 3.1 for details).

**Concept Frequency from Captions.** For efficient concept searches, we pre-index all captions from the pretraining datasets, *i.e.*, construct a mapping from concepts to captions. We first use part-of-speech tagging to isolate common and proper nouns, and lemmatize them with SpaCy [58] (lemmatization helps standardize verbs, enabling the estimation of their frequencies too [65]). These lemmatized terms are then cataloged in inverted unigram dictionaries, mapping each term to all sample indices in the pretraining dataset containing that term. To determine the frequency of a concept, we examine the concept's unigrams within these dictionaries. For multi-word concepts, we split them into their constituent unigrams, and then independently search for all unigrams before intersecting their hit lists to get a list of matched sample indices. The frequency of the concept in text captions is the count of these intersecting sample indices. This algorithm hence allows scalable $\mathcal{O}(1)$ search with respect to the number of captions for any concept in pretraining dataset captions.

**Concept Frequency from Images.** Unlike text captions, we do not have a finite vocabulary for pre-indexing pretraining images. Instead, we collect all the $4,029$ downstream concepts and verify their presence in images using a pretrained image tagging model. We tested various open-vocabulary object detectors, multi-tagging models and image-text matching models, for this concept tagging task. We found that RAM++ [59]—an open-set tagging model that tags images based on a predefined list of concept descriptions, in a multi-label manner—performs the best. We automatically consider the relationship between concepts (like synonyms) and concept hierarchies [84], since RAM++ uses descriptions generated by a language model (Appx. I.3) for each concept, to tag each image with certain concepts. This approach generates a list of pretraining images, each tagged with whether the downstream concepts are present or not, from which we can compute concept frequencies (Appx. I).

**Image-Text Matched Concept Frequencies.** Finally, we combine the frequencies obtained from both text and image searches to compute *matched image-text frequencies*. This involves identifying pretraining samples where both the image and its associated caption correspond to the concept. By intersecting the lists from our image and text searches, we determine the count of samples that align in both modalities, offering a comprehensive view of concept representation in the pretraining datasets. This step is necessary as we observed significant image-text misalignment between concepts in pretraining datasets (see Tab. 3), hence captions may not reflect what is present in the image and vice-versa. This behaviour has also been alluded to in prior work investigating data curation strategies [78, 77, 134, 89]. We provide a more detailed analysis of image-text misalignment in Sec. 5.

## 3 Comparing Pretraining Frequency & "Zero-Shot" Performance

Equipped with frequency estimates for downstream concepts, we now establish the relationship between image-text-matched pretraining concept frequencies and zero-shot performance across classification, retrieval, and generation tasks. We first detail our setup and then discuss key results.

Table 1: Pretraining and downstream datasets used in Image-Text (CLIP) experiments.

| Dataset Type | Datasets |
|---|---|
| Pretraining | CC-3M [115]   CC-12M [28]   YFCC-15M [123]   LAION-400M [110] |
| Classification-Eval | ImageNet [36]   SUN397 [133]   UCF101 [116]   Caltech101 [45]   EuroSAT [56]   CUB [131]
Caltech256 [50]   Flowers102 [90]   DTD [32]   Birdsnap [16]   Food101 [21]   Stanford-Cars [66]
FGVCAircraft [79]   Oxford-Pets [93]   Country211 [98]   CIFAR-10 [68]   CIFAR100 [68] |
| Retrieval-Eval | Flickr-1K [138]   COCO-5K [75] |

Table 2: Models used in text-to-image (T2I) experiments.

| Category | Models |
|---|---|
| Models | M-Vader [15]   DeepFloyd-IF-M [9]   DeepFloyd-IF-L [9]   DeepFloyd-IF-XL [9]
GigaGAN [63]   DALL·E Mini [35]   DALL.E Mega [35]   Promptist+SD-v1.4 [53]
Dreamlike-Diffusion-v1.0 [2]   Dreamlike Photoreal v2.0 [3]   OpenJourney-v1 [4]   OpenJourney-v2 [5]
SD-Safe-Max [104]   SD-Safe-Medium [104]   SD-Safe-Strong [104]   SD-Safe-Weak [104]
SD-v1.4 [104]   SD-v1.5 [104]   SD-v2-Base [104]   SD-v2-1-base [104]
Vintedois-Diffusion-v0.1 [7]   minDALL.E [105]   Lexica-SD-v1.5 [1]   Redshift-Diffusion [6] |

## 3.1 Experimental Setup

We analyze two classes of multimodal models: Image-Text and Text-to-Image. For both, we detail the pretraining and testing datasets, along with their associated evaluation parameters.

### 3.1.1 Image-Text (CLIP) Models

**Datasets.** We use 4 pretraining, 2 downstream retrieval, and 17 downstream classification datasets, covering a broad spectrum of objects, scenes, camera-types, and fine-grained distinctions (see Tab. 1).

**Note on Pretraining Dataset Diversity.** Each analyzed pretraining dataset significantly differs in data collection, filtering, and cleaning operations. CC-3M [115], originally intended to be used for training image captioning models, explicitly has no real-world entities or proper nouns present, and is cleaned only to have common nouns in its captions. CC-12M [28] and YFCC-15M [123], collected from Flickr, have user-provided metadata. Finally, LAION-400M [110] and LAION-Aesthetics [111] contain raw images downloaded from Common-Crawl [101] with alt-texts as the captions, which can be inherently very noisy as they are uploaded by non-expert humans as a placeholder for images.

**Models.** We test CLIP [98] models with both ResNet [54] and Vision Transformer [37] architecture, with ViT-B-16 [87] and RN50 [49, 88] trained on CC-3M and CC-12M, ViT-B-16, RN50, and RN101 [61] trained on YFCC-15M, and ViT-B-16, ViT-B-32, and ViT-L-14 trained on LAION400M [110]. We follow `open_clip` [61], `slip` [87] and `cyclip` [49] for our implementation.

**Prompting.** For zero-shot classification, we experiment with three prompting strategies: {classname} only, "A photo of a {classname}" and prompt-ensembles [98], which averages over 80 different prompt variations of {classname}. For retrieval, we use the image or the caption as input corresponding to I2T (image-to-text) or T2I (text-to-image) retrieval respectively.

**Metrics.** We compute mean accuracy for classification tasks [98]. For retrieval, we measure Recall@1, Recall@5, and Recall@10 for both text-to-image and image-to-text retrieval tasks [98].

### 3.1.2 Text-to-Image Models

**Datasets.** Our pretraining dataset is LAION-Aesthetics [111], with downstream evaluations done on subsampled versions of eight datasets: CUB200 [131], Daily-DALLE [34], Detection [31], Parti-Prompts [140], DrawBench [106], COCO-Base [75], Relational Understanding [33] and Winoground [124]. Please refer to HEIM [72] for more details on the evaluation datasets.

**Models.** We evaluate 24 T2I models, detailed in Tab. 2. Their sizes range from 0.4B parameters (DeepFloyd-IF-M [9] and DALL·E Mini [35]) to 4.3B parameters (DeepFloyd-IF-XL [9]). We include various Stable Diffusion models [104] as well as variants tuned for specific visual styles [6, 4, 5].

**Prompting.** Text prompts from the evaluation datasets are used directly to generate images, with 4 image samples generated for each prompt.

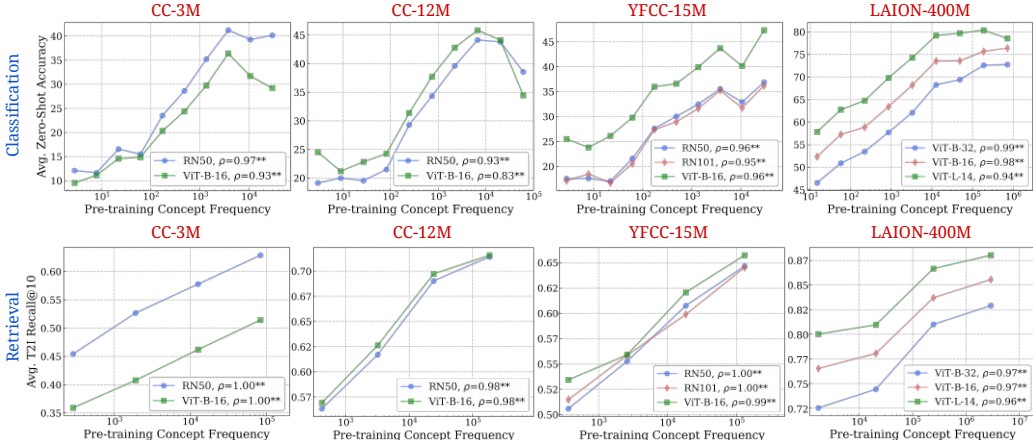

Figure 2: **Log-linear relationships between concept frequency and CLIP zero-shot performance.** Across all tested architectures (RN50, RN101, ViT-B-32, ViT-B-16, ViT-L-14) and pretraining datasets (CC-3M, CC-12M, YFCC-15M, LAION-400M), we observe a consistent linear relationship between CLIP's zero-shot performance on a concept and the log-scaled pretraining concept frequency. This trend holds for both zero-shot classification (results averaged across 17 datasets) and image-text retrieval (results averaged across 2 datasets). ** indicates that the result is significant ($p < 0.05$ with a two-tailed t-test [118]), and thus we show Pearson correlation ($\rho$) [73] as well.

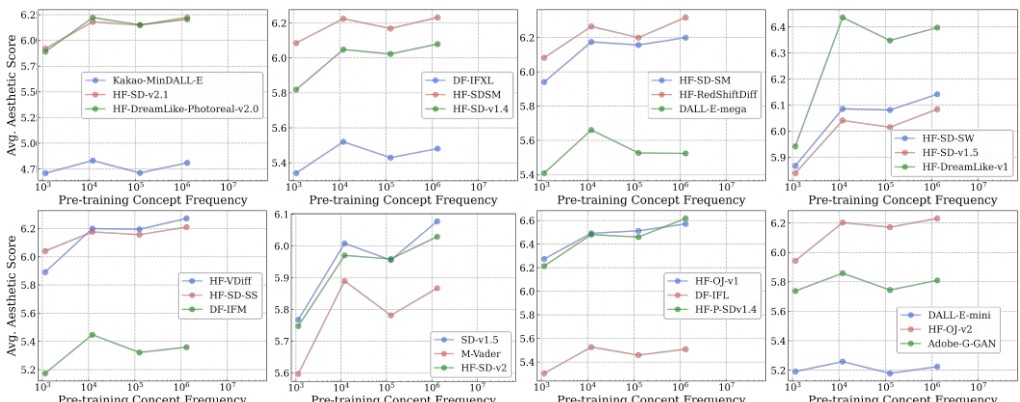

Figure 3: **Log-linear relationships between concept frequency and T2I aesthetic scores.** Across all tested T2I models pretrained on LAION-Aesthetics, we observe a consistent linear relationship between aesthetic score (averaged across 8 datasets) on a concept and the log-scaled concept frequency.

**Metrics.** Evaluation consists of image-text alignment and aesthetic scores. For automated metrics [72], we use expected and max CLIP-score [57] to measure image-text alignment along with expected and max aesthetics-score [110] to measure aesthetics. To verify reliability of automated metrics, we compare them with human-rated scores (measured on a 5-point grading scale) for both image-text alignment and aesthetics [72]. To supplement the human-rated scores provided by HEIM [72], we confirm our findings by performing our own small-scale human evaluation (Appx. C).

## 3.2 Result: Pretraining Concept Frequency is Predictive of "Zero-Shot" Performance

We now probe the impact of pretraining concept frequency on "zero-shot" performance of models. Our main results, across various tasks and model types, are shown in Figs. 2 and 3.

**Understanding the Plots.** The plots in the main paper present text-image (CLIP) models' zero-shot classification results using accuracy and text-to-image retrieval performance using Recall@10. Similarly, we present T2I generative models' performance on image generation tasks using the expected aesthetics score. For the other aforementioned metrics for retrieval as well as other automated generation metrics along with human-rated scores, we find that they show similar trends,

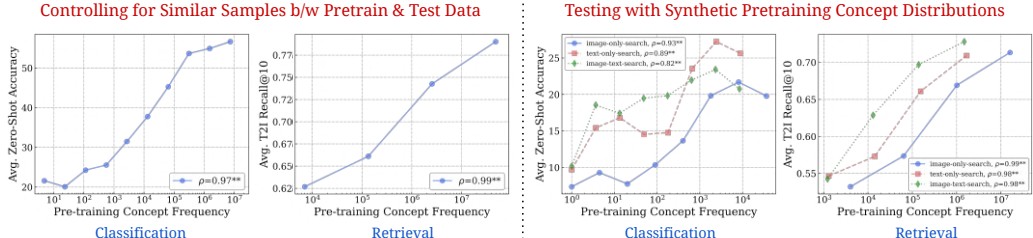

Figure 4: **Stress-testing the log-linear scaling trends.** We provide further evidence for the log-linear relationship between performance and concept frequency, across different scenarios: (*left*) we control for "similarity" between downstream test sets and pretraining datasets, and (*right*) we conduct experiments on an entirely synthetic pretraining distribution with no real-world images or captions.

and we provide them for reference in Apps. B and C. For clarity, the data presentation is simplified from scatter plots to a cohesive line, similar to work from Kandpal et al. [62] and Razeghi et al. [102]. The x-axis is log-scaled, and performance metrics are averaged within bins along this axis for ease-of-visualization of the log-linear correlation. We removed bins containing very few concepts per bin by standard IQR removal [132] following Kandpal et al. [62]. We additionally compute Pearson correlation $\rho$ [73] for each line and provide significance results based on a two-tailed t-test [118].

*Key Finding: Log-linear scaling between concept frequency and zero-shot performance.* Across all the 16 different plots, we observe a clear log-linear relationship between pretraining concept frequency and zero-shot performance. These plots vary in (i) model type (discriminative vs. generative), (ii) task (classification vs. retrieval), (iii) model architecture and parameter count, (iv) pretraining dataset (curation methods and scales), (v) evaluation metrics, (vi) prompting strategies, and (vii) concept frequencies isolated only from image or text caption (additional experiments for (v) are presented in Apps. B and C, for (vi) are presented in Appx. A, and for (vii) are presented in Appx. D). The observed log-linear scaling trend persists *across all seven presented dimensions*. In some plots, we notice a slight dip at the high-frequency concepts—we analyse this in greater detail in Appx. L. Thus, taken together, our results reveal data-hungry learning, *i.e*, a lack in current multimodal models' ability to learn concepts from pretraining datasets in a sample-efficient manner.

## 4 Stress-Testing Frequency-Performance Trends with Distributional Controls

In this section, we perform two control experiments to account for different confounding distributional factors of pretraining datasets, to ensure the robustness of our log-linear frequency-performance scaling trends: (1) we control for sample-level similarity in distribution between pretraining and evaluation datasets [137, 81], and (2) we investigate effects of pretraining data with radically different controlled concept distributions, with entirely synthetically-generated image-text pairs [52].

### 4.1 Controlling for Similar Samples in Pretraining and Downstream Data

**Motivation.** Prior work has suggested that sample-level similarity between pretraining and downstream datasets impacts model performance [62, 81, 137, 102]. This leaves open the possibility that our frequency-performance results are simply an artifact of this factor, *i.e.*, as concept frequency increases, it is likely that the pretraining dataset also contains more similar samples to the test sets. We hence investigate the frequency-performance trends when controlling for sample-level similarity.

**Setup.** We use the LAION-200M [10] dataset for this experiment. We first verify that a CLIP-ViT-B-32 model pretrained on the LAION-200M dataset (used to study sample similarity in prior work [81]) exhibits a similar log-linear trend between concept frequency and zero-shot performance. Then, we use the `near_pruning` method from Mayilvahanan et al. [81] to eliminate 50 million samples most similar to the test sets from the pretraining LAION-200M dataset. We provide details for this in Appx. G.1. This procedure removes the most similar samples between pretraining and test sets. We verify that this procedure influences the performance of the model drastically across our aggregate classification and retrieval tasks respectively, replicating the findings of Mayilvahanan et al. [81].

**Key Finding:** *Concept Frequency is still Predictive of Performance.* We repeat our analysis on models trained with this controlled pretraining dataset with 150M samples, and report results on the

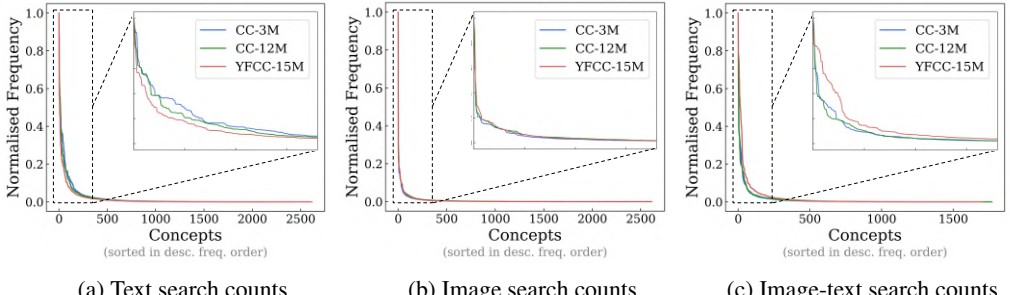

|(a) Text search counts|(b) Image search counts|(c) Image-text search counts|

Figure 5: **Concept distribution of pre-training datasets is highly long-tailed.** We showcase the distribution of pretraining frequencies of all concepts aggregated across all 17 of our downstream classification datasets. Across all the pretraining datasets, we observe very heavy tails. We normalize the concept frequencies and remove concepts with 0 counts for improved readability of the plots.

same downstream classification and retrieval datasets, in Fig. 4 (left). Despite removing the most similar samples between pretraining and test sets, we still consistently observe a clear log-linear relationship between pretraining frequency of test set concepts and zero-shot model performance.

**Conclusion.** This analysis reaffirms that, despite removing pretraining samples closely related to the downstream evaluation datasets, the log-linear relationship between concept frequency and zero-shot performance persists. Note that this is despite substantial decreases in absolute performance, highlighting the robustness of concept frequency as a performance indicator for CLIP models.

## 4.2  Testing Generalization to Purely Synthetic Concept and Data Distributions

**Motivation.** Sampling across real-world data might not result in significant differences in concept distribution, as we will later show in Sec. 5. Hence, we repeat our analysis on a synthetic dataset designed with an explicitly different concept distribution [52]. This evaluation aims to understand if pretraining concept frequency remains a significant performance predictor within a synthetic concept distribution, generalizing even for models pretrained on entirely synthetic images and text captions.

**Setup.** The SynthCI-30M dataset [52] introduces a novel concept distribution, generating 30 million synthetic image-text pairs. Utilizing their publicly available data and models, we explore the relationship between concept frequency and model performance in this purely synthetic data regime.

**Key Finding:** *Concept Frequency is still Predictive of Performance.* We report results for models pretrained with the controlled SynthCI-30M dataset in Fig. 4 (right). We still consistently observe a clear log-linear relationship between concept frequency and zero-shot model performance.

**Conclusion.** This consistency highlights that concept frequency is a robust indicator of model performance, extending even to entirely synthetic datasets and pretraining concept distributions.

## 5  Additional Insights from Pretraining Concept Frequencies

**Finding 1:** *Pretraining Datasets Exhibit Long-tailed Concept Distributions.* Our analysis in Fig. 5 reveals an extremely long-tailed distribution of concept frequencies in pretraining datasets, with over two-thirds of concepts occurring at almost negligible frequencies relative to the size of the datasets (we highlight the "head" part of this distribution in boxes). Our observations extend the findings of past work that have noted the long-tailed distribution of large-scale language datasets [14, 26, 94, 146]. As we observed with the log-linear trend, this distribution directly reflects disparities in performance.

**Finding 2:** *Misalignment Between Concepts in Image-Text Pairs.* Our concept frequency estimation pipeline enables us to investigate the alignment of concepts within paired pretraining image-text data. Perfect image-text alignment is defined as every image-text pair containing the same concepts. Previous studies have qualitatively discussed the problem of misalignment in large image-text datasets [77, 134, 78]. Our analysis enables us to quantify this *misalignment degree*—for each image-text pair in the pretraining dataset, we find concepts that are matched to the image and the text caption independently. If there are no intersecting concepts from the independent image and text hits, we mark that pair as misaligned (detailed algorithm shown in Appx. K). Tab. 3 shows the high degree of misalignment in all image-text pairs (5−36%). To the best of our knowledge, this is the first

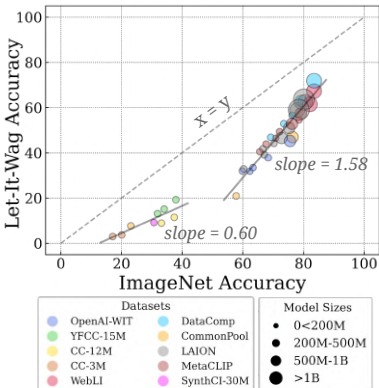

| Dataset/ Misalignment | Number of Misaligned pairs | Misalignment Degree (%) |
|---|---|---|
| **CC-3M** | 557,683 | 16.81% |
| **CC-12M** | 2,143,784 | 17.25% |
| **YFCC-15M** | 5,409,248 | 36.48% |
| **LAION-A** | 23,104,076 | 14.34% |
| **LAION-400M** | 21,996,097 | 5.31% |

Table 3: For each pretraining dataset, we present the number of misaligned image-text pairs and the *misalignment degree*: fraction of misalignment pairs.

| Correlations | CC-3M | CC-12M | YFCC-15M | LAION-400M |
|---|---|---|---|---|
| **CC-3M** | 1.00 | 0.79 | 0.96 | 0.63 |
| **CC-12M** | – | 1.00 | 0.97 | 0.74 |
| **YFCC-15M** | – | – | 1.00 | 0.76 |
| **LAION-400M** | – | – | – | 1.00 |

Table 4: We compute correlation in concept frequency across pretraining datasets, observing strong correlations, despite major differences in scale and curation.

Figure 6: **Large-drops in accuracy on "*Let It Wag!*".** Across 40 tested CLIP models, we note large performance drops compared to ImageNet. Further, the performance gap seems to decrease for high-capacity models as demonstrated by larger positive slope (1.58) for those models.

attempt to quantify the misalignment degree in pretraining image-text datasets explicitly. We release the precise misaligned image-text samples from pretraining datasets to enable better data curation.

**Finding 3:** *Concept Frequencies Across Datasets are Correlated.* Despite vast differences in the size (3M-400M samples) and curation strategies of the pretraining datasets, we discovered a surprisingly high correlation in concept frequencies across them (see Tab. 4). This suggests that the internet, as the common source of these datasets, naturally exhibits a long-tailed distribution, influencing any dataset derived from it to display similar long-tailed behavior also. This result inspired "*Let It Wag!*".

## 6 Testing the Tail: *Let It Wag!*

**Motivation.** In the previous sections, we identified a consistent long-tailed concept distribution across pretraining datasets, highlighting the scarcity of certain concepts on the web. This observation forms the basis of our hypothesis that models likely underperform when tested against data distributions that are heavily long-tailed. To test this, we carefully curate 290 concepts identified as the least frequent across all pretraining datasets. This includes concepts like eggnog, wormsnake, and tropical kingbird. We then use these concepts to create an evaluation dataset, "*Let It Wag!*".

**Dataset Details.** The *"Let It Wag!"* classification dataset comprises 130K test samples downloaded from the web using the method of Prabhu et al. [96]. The test samples are evenly distributed across 290 categories of long-tailed concepts. From the list of curated concepts, we download test set images, deduplicate them, remove outliers, and finally manually clean and hand-verify the class labels.

**Analysis Details.** We run both classification and image generation experiments on *"Let It Wag!"*. For classification, we evaluate 40 text-image (CLIP) models on the *"Let It Wag!"* classification dataset, using an ensemble of 80 prompts from Radford et al. [98]. For the generation task, we utilize SD-XL [95], SD-v2 [104], and Dreamlike-Photoreal-v2.0 [3], to generate images for the long-tailed concepts. For each model, we run 50 diffusion steps, maintaining default settings for all other parameters.

**Text-Image Classification Results.** We showcase the results of our long-tailed classification task in Fig. 6—we plot results of all models on both *"Let It Wag!"* (y-axis) and ImageNet (x-axis). We observe that all models underperform by large margins on the long-tailed *"Let It Wag!"* dataset (upto 20% lower absolute accuracies compared to ImageNet). This performance drop-off generalises across all model scales and 10 different pretraining data distributions, reinforcing the notion that all web-sourced pretraining datasets are inherently constrained to be long-tailed. With that said, note that the higher capacity models (fitted line with slope=1.58 in Fig. 6) seem to be closing the gap to ImageNet performance, indicating improved performance on the long-tailed concepts.

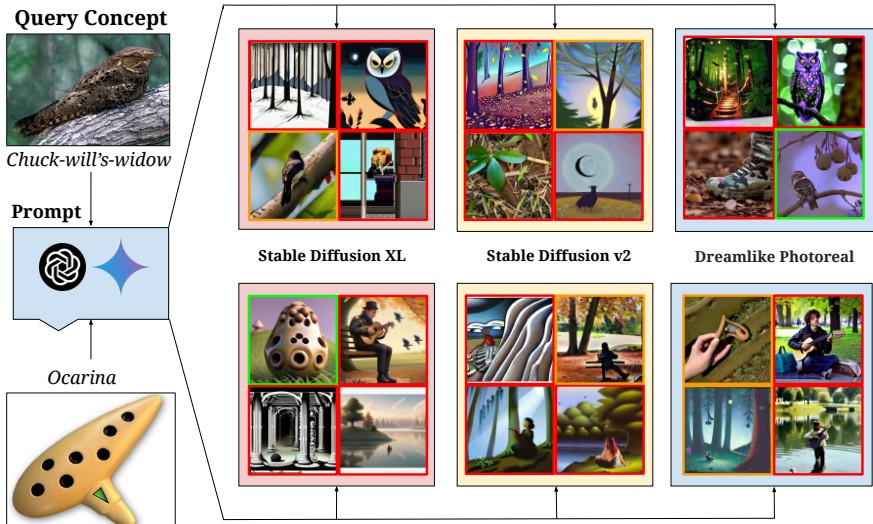

Figure 7: **Qualitative results on *"Let It Wag!"* concepts demonstrate failure cases of T2I models on the long-tail.** We created 4 prompts for each concept using Gemini [121] and GPT-4 [12] which are fed to 3 Stable Diffusion [104] models. Generations with red border are incorrect, green border are correct and yellow border are ambiguous. Despite advances in high-fidelity image generation, there is large scope for improvement for such long-tail concepts (quantitative results in Appx. N.1).

**T2I Generation Results.** We provide a qualitative analysis on image generation for assessing T2I models on the rare *"Let It Wag!"* concepts in Fig. 7. For enhancing image diversity, we generate prompts using Gemini [121] (top row of generated images) and GPT-4 [12] (bottom row of generated images). Green borders represent correct generations, red borders represent incorrect generations and yellow borders represent ambiguous generations. While descriptive prompting generally aids in improving the quality of generated images [53], we still observe T2I models failing to comprehend and accurately represent many concepts in our *"Let It Wag!"* dataset. Some failure cases involve misrepresenting activities (such as Pizza Tossing or Cricket Bowling as shown in Fig. 29), generating the wrong concept (Chuck-will's-widow as shown in Fig. 7 (top)), as well as not comprehending the concept at all (Ocarina in Fig. 7 (bottom)). We hence show that Stable Diffusion models are prone to the long tail qualitatively—we also provide quantitative results in Appx. N.1.

**Conclusion.** Across both the classification and generation experiments, we have showcased that current multimodal models predictably underperform, regardless of their model scale or pretraining datasets. This suggests a need for better strategies for sample-efficient learning on the long-tail.

## 7 Related Work

We discuss the most relevant prior works to ours here, and defer an extended literature review to Appx. F. Past works [98, 47, 88, 43, 76, 39, 82, 81, 67] have highlighted the importance of pretraining data for improved downstream model performance. Fang et al. [43] demonstrated that pretraining data diversity is key to CLIP's strong out-of-distribution generalisation. Nguyen et al. [88] extended this analysis to show that differences in data distributions can change model performance, enabling effective data mixing strategies for pretraining. Mayilvahanan et al. [81] complemented these works by showing that CLIP's performance is correlated with the similarity between pretraining and test datasets. Our findings further pinpoint that the frequency of concept occurrences is a key indicator of performance. This complements existing work in areas like question-answering [62] and numerical reasoning [102] in LLMs. Concurrent to our work, Parashar et al. [92] also explore the problem of long-tailed concepts in LAION-2B and how it affects CLIP performance, supporting our findings. In contrast to their work, our demonstration that the long tail yields a log-linear trend, explicitly indicates exponential sample inefficiency in pretrained multimodal models. Additionally, contrary to their work, we index both image and text modalities, as well as span across several scales of diverse pretraining datasets. Our frequency estimation procedure on both texts and images independently, enables us to provide a more finer-grained analysis of pretraining datasets than previously studied in the literature, like (1) quantifying the misalignment between images and text captions, (2) assessing

the similarity of the different pretraining data concept distributions, and (3) doing a number of control experiments to thoroughly stress-test the robustness of our log-linear scaling results.

# 8 Conclusion

In this work, we studied 5 pretraining datasets of 34 multimodal models, analyzing the distribution and composition of concepts within them, generating over 300GB of data artifacts that we publicly release. Our findings reveal that across concepts, significant improvements in zero-shot performance require exponentially more data, following a sample-inefficient log-linear scaling trend. This pattern persists despite controlling for similarities between pretraining and downstream datasets or even when testing models on entirely synthetic data distributions. Further, all tested models consistently underperformed on the *"Let it Wag!"* dataset, which we systematically constructed from our findings to test for long-tail concepts. This underlines a critical reassessment of what "zero-shot" generalization entails for multimodal models, highlighting the limitations in their current generalization capabilities.

## Acknowledgements

The authors would like to thank (in alphabetic order of first name): Gyungin Shin, Jonathan Roberts, Karsten Roth, Mehdi Cherti, Prasanna Mayilvahanan, Shyamgopal Karthik, Thao Nguyen and Vlad Bogolin for helpful feedback and providing access to various resources throughout the project. YS would like to thank Wieland Brendel, Nicholas Carlini, Daphne Ippolito, Katherine Lee, Matthew Jagielski, and Milad Nasr. AP is funded by Meta AI Grant No. DFR05540. VU and YS thank the International Max Planck Research School for Intelligent Systems (IMPRS-IS). VU also thanks the European Laboratory for Learning and Intelligent Systems (ELLIS) PhD program for support. VU was supported by a Google PhD Fellowship in Machine Intelligence. PT thanks the Royal Academy of Engineering for their support. AB acknowledges the funding from the KAUST Office of Sponsored Research (OSR-CRG2021-4648) and the support from Google Cloud through the Google Gemma 2 Academic Program GCP Credit Award. SA is supported by a Newton Trust Grant. MB acknowledges financial support via the Open Philanthropy Foundation funded by the Good Ventures Foundation. This work was supported by the German Research Foundation (DFG): SFB 1233, Robust Vision: Inference Principles and Neural Mechanisms, TP4, project number: 276693517 and the UKRI grant: Turing AI Fellowship EP/W002981/1. MB is a member of the Machine Learning Cluster of Excellence, funded by the Deutsche Forschungsgemeinschaft (DFG, German Research Foundation) under Germany's Excellence Strategy – EXC number 2064/1 – Project number 390727645.

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

# Part I

# Appendix

## Table of Contents

# A Concept Frequency is Predictive of Performance Across Prompting Strategies

We extend the zero-shot classification results from Fig. 2 in Fig. 8 with two different prompting strategies: the results in the main paper used the {classname} only as the prompts, here we showcase both (1) "A photo of a {classname}" prompting and (2) 80 prompt ensembles as used by Radford et al [98]. We observe that ***the strong log-linear trend between concept frequency and zero-shot performance consistently holds across different prompting strategies***.

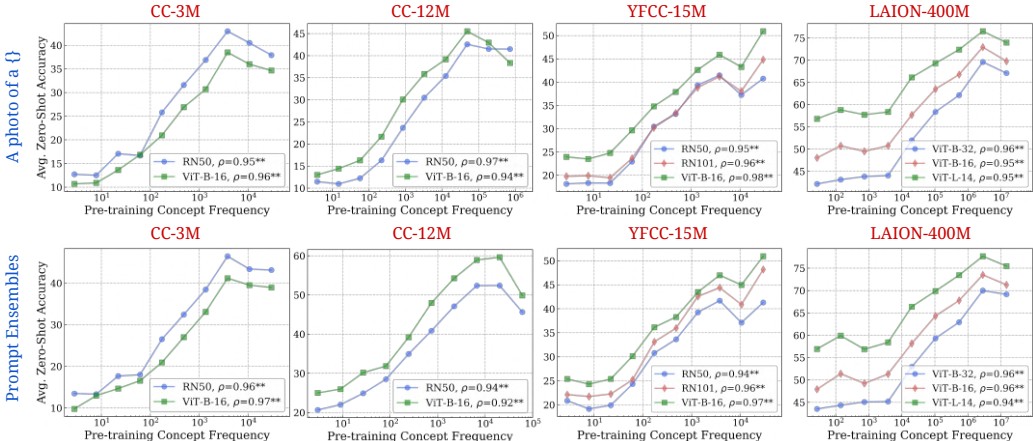

Figure 8: **Log-linear relationships between concept frequency and CLIP zero-shot performance.** Across all tested architectures (RN50, RN101, ViT-B-32, ViT-B-16, ViT-L-14) and pretraining datasets (CC-3M, CC-12M, YFCC-15M, LAION-400M), we observe a consistent linear relationship between CLIP's zero-shot classification accuracy on a concept and the log-scaled concept pretraining frequency. This trend holds for both "A photo of a {classname}" prompting style and 80 prompt ensembles [98]. ** indicates that the result is significant ($p < 0.05$ with a two-tailed t-test.), and thus we show Pearson correlation ($\rho$) as well.

# B Concept Frequency is Predictive of Performance Across Retrieval Metrics

We supplement Fig. 2 in the main paper, where we showed results with the text-to-image (I2T) recall@10 metric. In Figs. 9 and 10, we present results for the retrieval experiments across all six metrics: I2T-Recall@1, I2T-Recall@5, I2T-Recall@10, T2I-Recall@1, T2I-Recall@5, T2I-Recall@10. We observe that *the strong log-linear trend between concept frequency and zero-shot performance robustly holds across different retrieval metrics*.

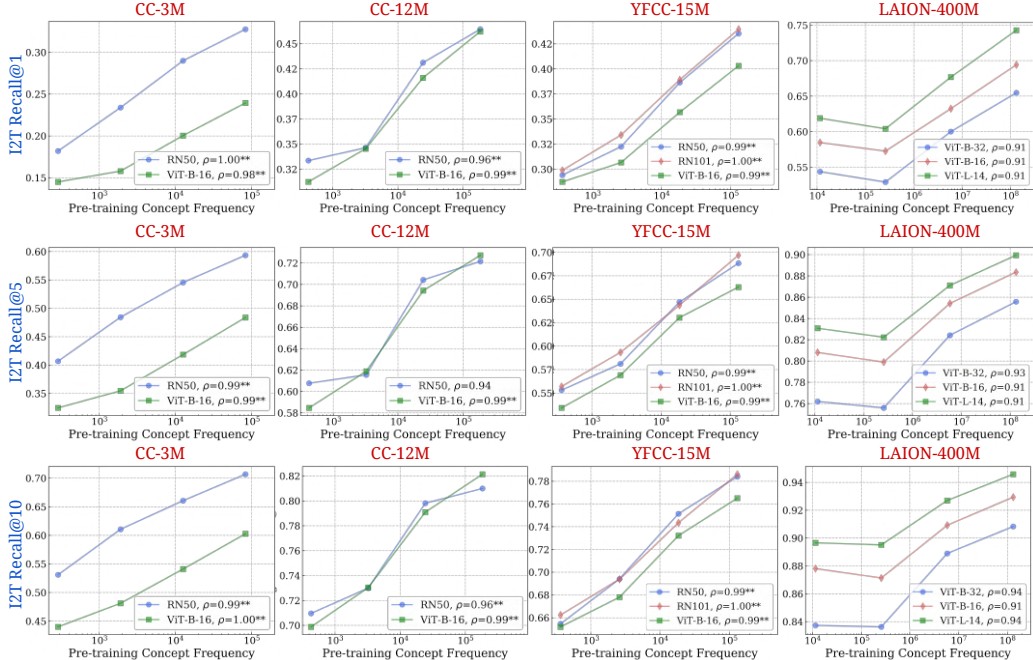

Figure 9: **Log-linear relationships between concept frequency and CLIP I2T retrieval performance.** Across all tested architectures (RN50, RN101, ViT-B-32, ViT-B-16, ViT-L-14) and pretraining datasets (CC-3M, CC-12M, YFCC-15M, LAION-400M), we observe a consistent linear relationship between CLIP's retrieval performance (measured using image-to-text metrics) on a concept and the log-scaled concept pretraining frequency. ** indicates that the result is significant ($p < 0.05$ with a two-tailed t-test.), and thus we show Pearson correlation ($\rho$) as well.

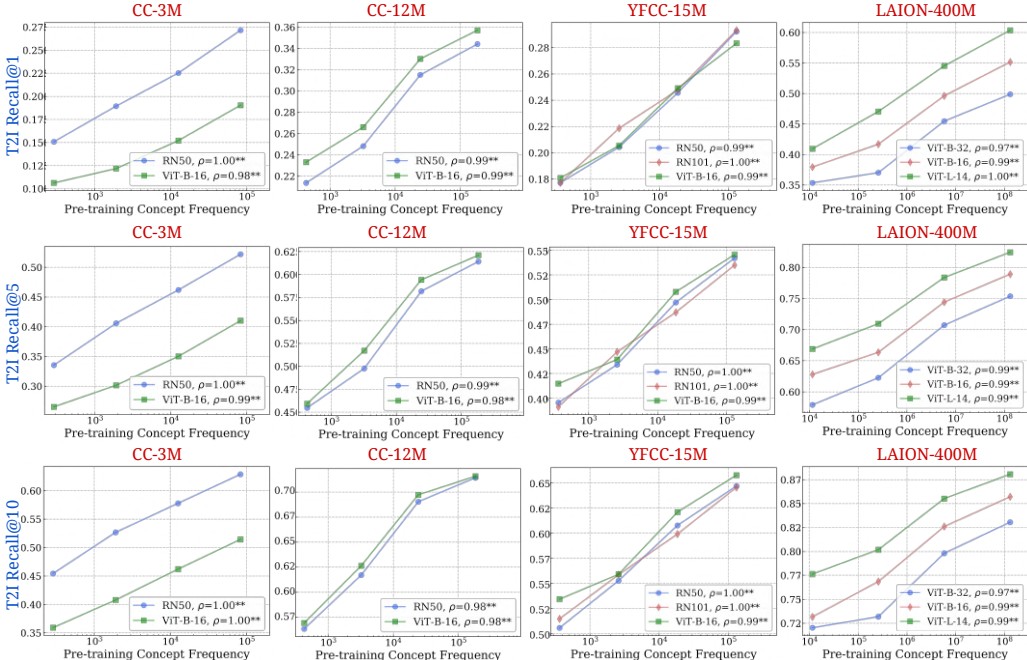

Figure 10: **Log-linear relationships between concept frequency and CLIP T2I retrieval performance.** Across all tested architectures (RN50, RN101, ViT-B-32, ViT-B-16, ViT-L-14) and pretraining datasets (CC-3M, CC-12M, YFCC-15M, LAION-400M), we observe a consistent linear relationship between CLIP's retrieval performance (measured using text-to-image metrics) on a concept and the log-scaled concept pretraining frequency. ** indicates that the result is significant ($p < 0.05$ with a two-tailed t-test.), and thus we show Pearson correlation ($\rho$) as well.

## C   Concept Frequency is Predictive of Performance for T2I Models

We extend the results from Fig. 3 with Figs. 11 to 15. As with Fig. 3, due to the high concept frequency, the scaling trend is slightly less pronounced. Furthermore, we do see inconsistency in the trends for the human-rated scores retrieved from HEIM [72], hence we perform our own small scale human evaluation to check them.

**Human Study with People Concepts.** Given the societal relevance [24], we decided to test Stable Diffusion [104] (v1.4) on generating public figures. We scraped 50,000 people from the "20230123-all" Wikidata JSON dump by filtering for entities listed as "human" [8], and scraped a reference image for the human study for each person if an image was available. After computing concept frequency from LAION-Aesthetics text captions (using suffix array [71]), we found that ≈10,000 people were present in the pretraining dataset. Note that to ensure the people's names were treated as separate words, we computed frequency for strings of the format " {entity} ". We then randomly sample 360 people (for which a reference image was available) normalized by frequency [23] for the human study. For generating images with Stable Diffusion, we used the prompt "headshot of {entity}", in order to specify to the model that "{entity}" is referring to the person named "{entity}" [51].

We assessed image-text alignment with a human study with 6 participants, where each participant was assigned 72 samples; for consistency, of the 360 total samples, we ensured 10% were assigned to 3 participants. Provided with a reference image, the participants were asked if the sample accurately depicts the prompt (see Fig. 16). Specifically, "Does the image accurately depict the above prompt?". Three choices were provided: "Yes" (score=1.), "Somewhat" (score=0.5), and "No" (score=0.). Accuracy was computed by averaging the scores.

As can be seen in Fig. 17, we observe a log-linear trend between concept frequency and zero-shot performance. Thus, we observe that *the log-linear trend between concept frequency and zero-shot performance consistently holds even for T2I models*.

**Note on Participant Acquisition.** Experiment participants, who volunteered for the study, provided informed consent. IRB approval was not obtained.

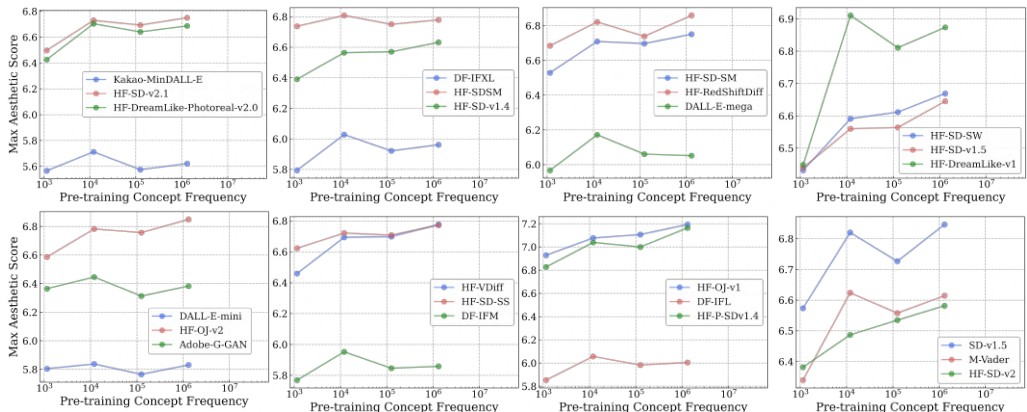

Figure 11: **Log-linear relationships between concept frequency and T2I Max aesthetic scores.** Across all tested models pretrained on the LAION-Aesthetics dataset, we observe a consistent linear relationship between T2I zero-shot performance on a concept and the log-scaled concept pretraining frequency.

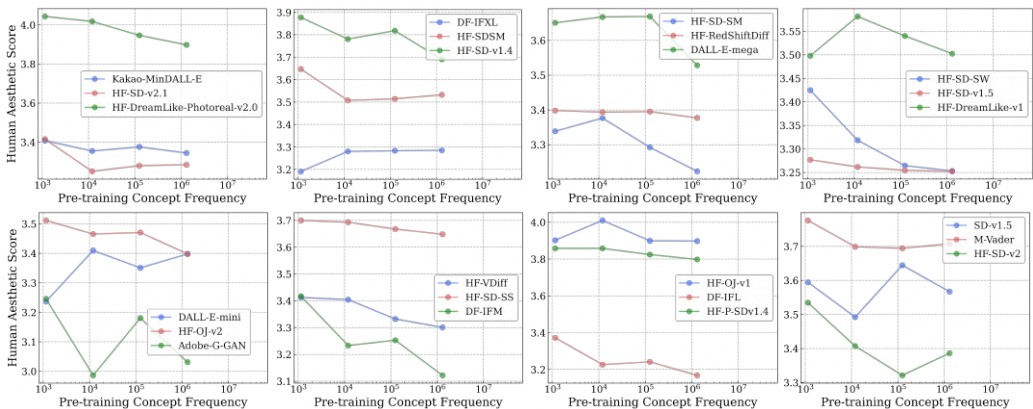

Figure 12: **Log-linear relationships between concept frequency and T2I human aesthetic scores.** Across all tested models pretrained on the LAION-Aesthetics dataset, we observe a consistent linear relationship between T2I zero-shot performance on a concept and the log-scaled concept pretraining frequency.

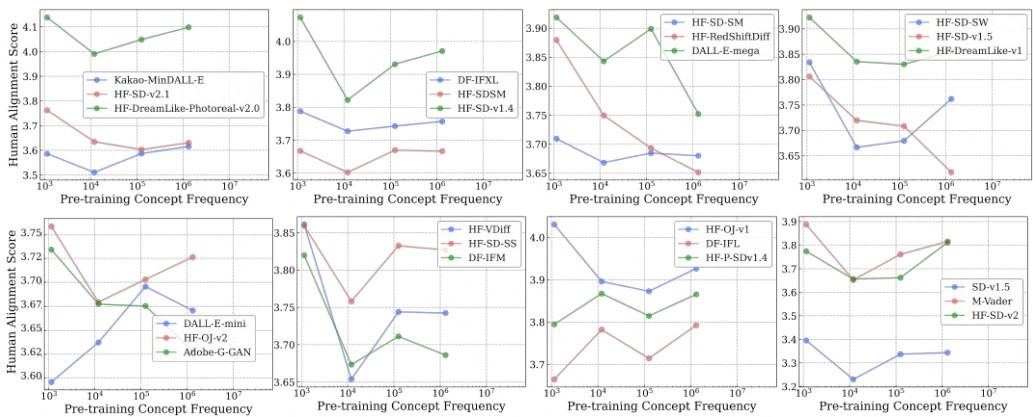

Figure 13: **Log-linear relationships between concept frequency and T2I human alignment scores.** Across all tested models pretrained on the LAION-Aesthetics dataset, we observe a consistent linear relationship between T2I zero-shot performance on a concept and the log-scaled concept pretraining frequency.

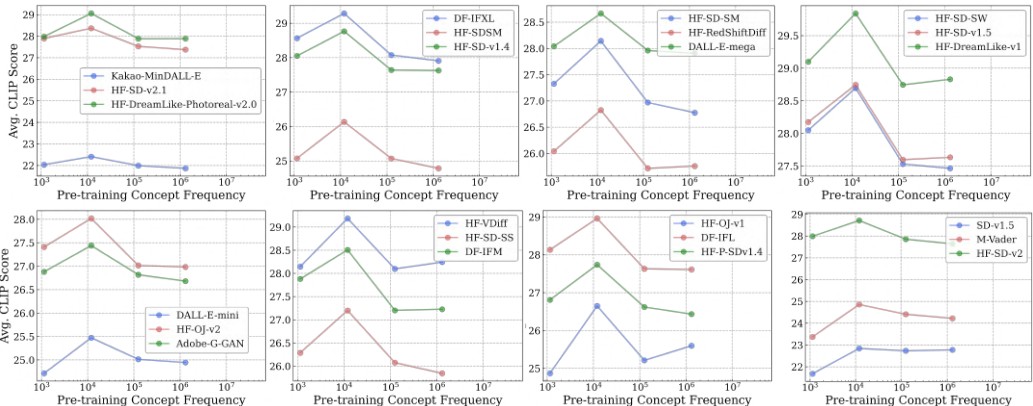

Figure 14: **Log-linear relationships between concept frequency and T2I Avg. CLIP scores.** Across all tested models pretrained on the LAION-Aesthetics dataset, we observe a consistent linear relationship between T2I zero-shot performance on a concept and the log-scaled concept pretraining frequency.

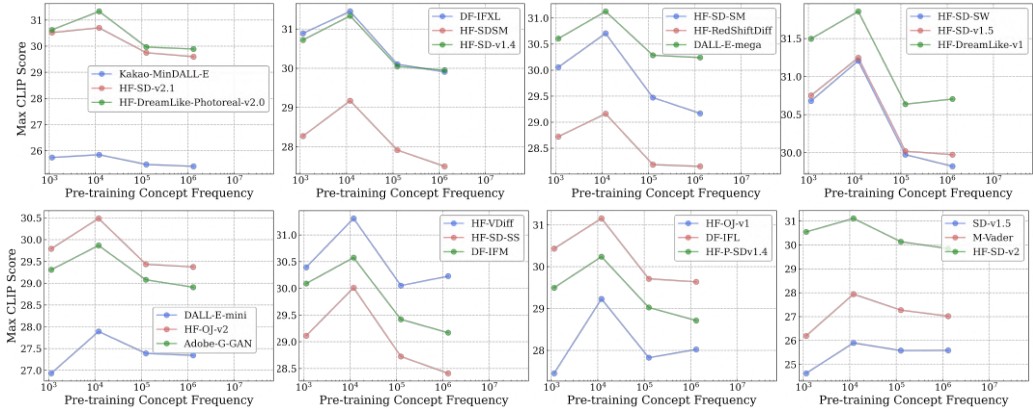

Figure 15: **Log-linear relationships between concept frequency and T2I Max CLIP scores.** Across all tested models pretrained on the LAION-Aesthetics dataset, we observe a consistent linear relationship between T2I zero-shot performance on a concept and the log-scaled concept pretraining frequency.

**Prompt: headshot of Berhaneyesus Demerew Souraphiel**

Image:                    Reference: Berhaneyesus Demerew Souraphiel

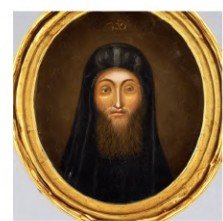 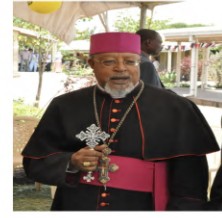

Does the image accurately depict the above prompt?

○ yes
○ somewhat
○ no
Submit

Figure 16: **User Interface for T2I human evaluation for text-image alignment for people concepts.** See Appx. C for further details.

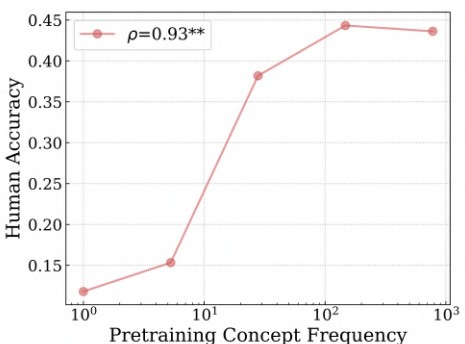

Figure 17: **Log-linear relationship between concept frequency and T2I human evaluation for text-image alignment for people concepts.** We observe a consistent linear relationship between T2I zero-shot performance on a concept and the log-scaled concept pretraining frequency.

## D Concept Frequency is Predictive of Performance across Concepts only from Image and Text Domains

In all the main performance-frequency plots we have presented until now, the concept frequencies were estimated using the intersection of the image-frequencies and the text-frequencies. Here, we showcase results with using them independently in Figs. 18 and 19 respectively. We note that both independent searching methods showcase log-linear trends as before confirming our main result. We observe that *the strong log-linear trend between concept frequency and zero-shot performance robustly holds across concepts derived from image and text domains independently as well*.

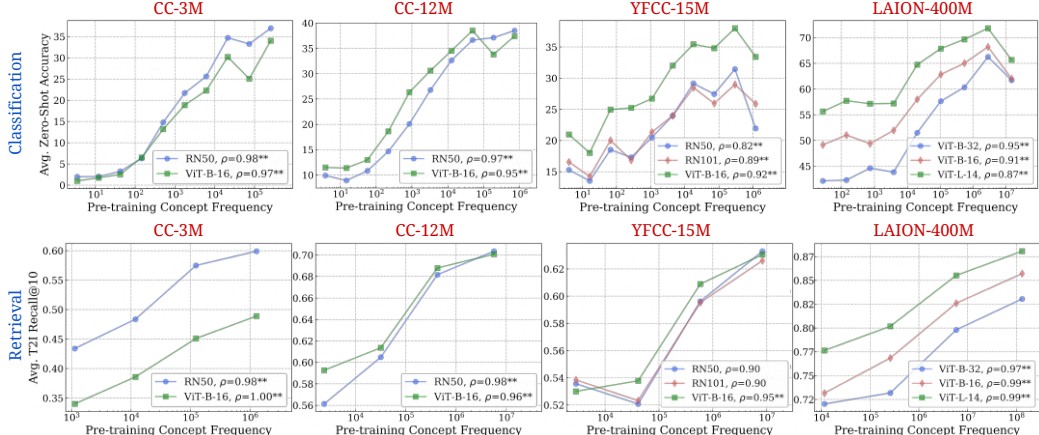

Figure 18: **Log-linear relationships between image concept frequency and CLIP performance.** Across all tested architectures (RN50, RN101, ViT-B-32, ViT-B-16, ViT-L-14) and pretraining datasets (CC-3M, CC-12M, YFCC-15M, LAION-400M), we observe a consistent linear relationship between CLIP's zero-shot accuracy and retrieval performance on a concept and the log-scaled concept pretraining frequency (searched using only pretraining images). ** indicates that the result is significant ($p < 0.05$ with a two-tailed t-test.), and thus we show Pearson correlation ($\rho$) as well.

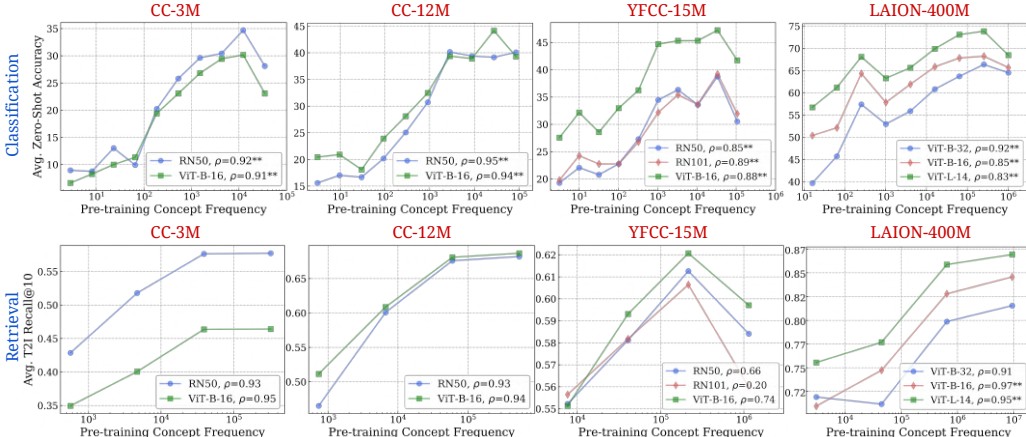

Figure 19: **Log-linear relationships between text concept frequency and CLIP performance.** Across all tested architectures (RN50, RN101, ViT-B-32, ViT-B-16, ViT-L-14) and pretraining datasets (CC-3M, CC-12M, YFCC-15M, LAION-400M), we observe a consistent linear relationship between CLIP's zero-shot accuracy and retrieval performance on a concept and the log-scaled concept pretraining frequency (searched using only pretraining text captions). ** indicates that the result is significant ($p < 0.05$ with a two-tailed t-test.), and thus we show Pearson correlation ($\rho$) as well.

# E   Generalization of findings to improved VLM training objectives

We believe our main conclusions of exponential data inefficiency should hold regardless of the model architecture and the training objective for any VLM. However, to test this thoroughly, we investigated two methods that have been empirically shown to improve generalization capabilities of CLIP models: CyCLIP [49] and SLIP [87]. We use 4 different models, each trained with either CyCLIP/SLIP on three different datasets—we then plot our main log-linear scaling results similar to Fig. 2 for CyCLIP and SLIP models—these plots are in Fig. 20. We observe for both SLIP and CyCLIP models, the log-linear scaling trends hold strong, with high Pearson correlation coefficients, further signifying the robustness of our main results. Hence, we emphasize that our main conclusions hold true even when considering multimodal models that explicitly introduce new training objectives with the aim of improving model generalization.

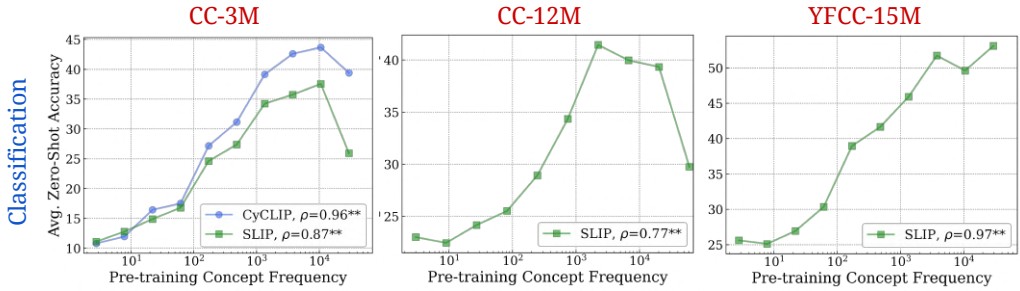

Figure 20: **Log-linear scaling trends for SLIP and CyCLIP models**

# F   Extended Related Work

Supplementing the discussion in the main paper, we further provide a broad overview of the surrounding literature within which our paper is mainly positioned.

**Effect of Pre-training Data on Downstream Data.** Several data-centric prior works [98, 47, 88, 43, 89, 76, 134, 135, 145, 117, 80, 99, 107, 108, 39, 27, 103] have highlighted the importance of pretraining data in affecting performance. Fang et al. [43] robustly demonstrated that pretraining data diversity is the key property underlying CLIP's strong out-of-distribution generalisation behaviour. Similarly, Berlot-Attwell et al. [17] showed that attribute diversity is crucial for compositional generalization [60], namely systematicity [46]. Nguyen et al. [88] extended the Fang et al. [43] analysis to show that differences in data distributions can predictably change model performance and that this behaviour can lead to effective data mixing strategies at pretraining time. Mayilvahanan et al. [81] complemented this research direction by showing that CLIP's performance is correlated with the similarity between training and test datasets. Udandarao et al. [129] further showed that the frequency of certain visual data-types in the LAION-2B dataset was roughly correlated to the performance of CLIP models in identifying visual data-types. McCoy et al. [82] introduced the teleological approach to understanding model generalization, showing that LLMs are more reliable on tasks and input/output instances that are more probable based on their pretraining datasets. Our findings further pinpoint that the frequency of concept occurrences is a key indicator of performance. This complements existing research in specific areas like question-answering [62] and numerical reasoning [102] in large language models, where high train-test set similarity does not fully account for observed performance levels [137]. Concurrent to our work, Parashar et al. [92] also explore the problem of long-tailed concepts in the LAION-2B dataset and how it affects performance of CLIP models, supporting our findings. In contrast to their work, we look at count separately in image and text modalities, as well as across pretraining sets, and do a number of control experiments to thoroughly test the robustness of our result. Further, our frequency estimation procedure on text captions and images independently enables us to provide a more fine-grained analysis of the pretraining data distribution like quantifying misalignment between images and texts, and assessing similarity of the different pretraining data concept distributions. Finally, our demonstration that the long tail yields a log-linear trend explicitly indicates exponential sample inefficiency in large-scale pretrained models.

**Detailed Differences and Contributions compared to Parashar et al. [92].** We emphasise that we complement this prior work and point out that our work comprehensively tests the strength of the log-linear scaling trend across several datasets, spanning varying levels of data curation and dataset sizes. Further, we note that in Parashar et al. [92], the estimated frequencies are computed using only the text captions of LAION-2B. These estimated frequencies are then used as the canonical frequencies for plotting the performance-frequency curves for all the tested models (despite these models being trained on different pretraining datasets other than LAION-2B). Our work strongly showcases why this apparent asymmetry in their frequency estimation methodology should work—from Tab. 4, we show that different VLM pretraining datasets are strongly correlated in their concept distributions. Hence, in spite of Parashar et al. [92] using only LAION-2B as their source dataset for frequency estimation, their results roughly hold true because of this strong correlation across pretraining datasets. Our methodology of incorporating both images and text captions when computing the frequency estimates is crucial for explaining this. Hence, we believe that our work comprehensively generalizes and explains the findings of prior work while also providing insights into the pretraining datasets (*e.g.*, misalignment degree and correlation of concept distributions in datasets).

**Data-centric analyses.** Our work also adds to the plethora of work that aims to understand and explore the composition of large-scale datasets, and uses data as a medium for improving downstream tasks. Prior work has noted the importance of data for improving model performance on a generalised set of tasks [47, 11, 41, 13, 114]. For instance, several works utilise retrieved and synthetic data for adapting foundation models on a broad set of downstream tasks [128, 55, 125, 22, 109, 144, 96]. Maini et al. [78] observed the existence of "text-centric" clusters in LAION-2B and measured its impact on downstream performance. Other work has seeked to target the misalignment problem that we quantified in Tab. 3 by explicit recaptioning of pretraining datasets [69, 29, 130, 141, 89, 18]. Further, studies have also shown that by better data pruning strategies, neural scaling laws can be made more efficient than a power-law [117, 10]. Prior work has also showcased that large-scale datasets suffer from extreme redundancy in concepts, and high degrees of toxic and biased content [40, 126]. Further research has showcased the downstream effects that such biases during pretraining induce

in state-of-the art models [20, 112, 19, 48]. Our work tackles the issue of long-tailed concepts in pretraining datasets, and shows that this is an important research direction to focus efforts on.

# G  Experimental Details

## G.1  Setup of Mayilvahanan et al. [81]

LAION-200M is a dataset obtained by deduplicating LAION-400M by pruning exact duplicates, near duplicates, and semantically similar samples within LAION-400M [10]. The control pretraining set is created by pruning 50 million highly similar samples from LAION in the order of decreasing perceptual similarity to datapoints in ImageNet-val set. We use the 150M pretraining set for obtaining the concept distribution. We evaluate the performance of a ViT-B-32 CLIP model trained on this dataset on our downstream tasks and present our analysis on those tasks.

# H  *Let It Wag!* Test Set

## H.1  Final Set of Concepts in *Let It Wag!*

Based on our frequency estimation pipeline from Sec. 2, we carefully curate 290 of the least frequent concepts across LAION-400M pretraining dataset (out of the $4,029$). We then remove all the concepts that have 0 counts to ensure that our final dataset consists of concepts that have been detected atleast once in LAION-400M, this method has also been used in Kandpal et al. [62] to ensure robustness to noise in the estimation process. We then add them as our set of concepts in *Let It Wag!*. A few example concepts from our final list are: {`Beechcraft_1900, Black_Rosy_Finch, Irish_Wolfhound, Japanese_Chin, Kentucky_Warbler, eastern_hog-nosed_snake, eel, eggnog, flatfish, isopod, kingsnake, ladle, lakeshore, letter_opener`}. We release our full concept list publicly here.

**High-level insights about long-tail concepts.** The broad categories of the most long-tailed concepts with a few examples for each are as follows (a majority of them are also highlighted in Figs. 28 to 31):

- *Birds:* Western Scrub Jay, Cassins Finch, Prairie Warbler, Red eyed Vireo, Veery
- *Animals:* flatworm, Tibetan Mastiff, Scottish Terrier, vine snake, newt
- *Aircrafts:* A300B4, A310, Falcon 900, DHC-8-300, MD-11
- *Objects:* guillotine, letter opener, ladle, dust jacket
- *Plants and fungi:* mexican aster, gyromitra, great masterwort, thorn apple, cape flower
- *Misc.:* consomme, stratified texture, eggnog

### Further statistics of *Let-It-Wag!*

We provide some further statistics of the test-set below.

- *Most frequent concepts:* partridge (count=9489), Bank Swallow (count=9489), eel (7907)
- *Least frequent concepts:* Red-necked Grebe (count=0), SR-20 aircraft (count=0), Globe-flower (count=0)
- *Median frequency of concepts:* 97.5
- *Mean frequency of concepts:* 1096.2

We also show the full histogram of concept frequencies for the 290 concepts in *Let-It-Wag!* in Fig. 21. From the histogram, it is evident that most of the concepts in *Let-It-Wag!* have frequency less than 2000. About half of the concepts in *Let-It-Wag!* (approx. 140) have a frequency less than 1000. Hence, this histogram sufficiently establishes *that our Let-It-Wag! dataset truly captures the long tail*.

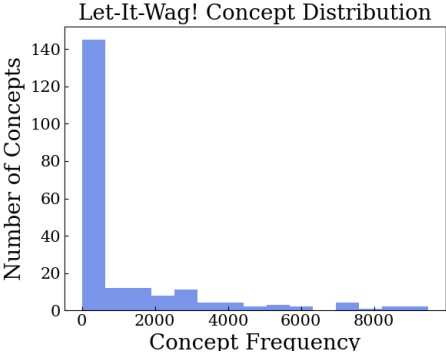

Figure 21: **Histogram of concept frequencies for *Let-It-Wag!* Dataset**

## H.2  *Let It Wag!*: Classification Test Set Curation

To ensure our "*Let It Wag!*" classification dataset is thoroughly cleaned and diverse, we follow a meticulous process consisting of several cleaning, filtering and verification steps:

**1. Diverse Sourcing:** We gather images from three different online sources—Flickr, DuckDuckGo, and Bing Search—to maximize the variety of our dataset while retaining very easy-to-classify images[3].

**2. Temporal Filtering:** We applied a filter to only retrieve images after January 2023 to minimize overlap with images used in the pretraining datasets of multimodal models. Note this helps mitigate but does not ensure that the overlap problem is resolved.

**3. Outlier Removal:** We employ a pre-trained InceptionNet [119] to remove outliers from the entire image pool. We do this by taking all pairwise cosine-similarities between all images in the pool, and removing the images that are in the bottom 5% of the similarity values[4].

**4. Initial De-duplication with an InceptionNet:** We employ a pre-trained InceptionNet [119] model to identify and remove duplicates. This step involves setting high thresholds for soft de-duplication (0.9 for common classes and 0.95 for fine-grained classes) to ensure only minor, precise exclusions. A threshold of 0.9/0.95 means that we consider images to be duplicates if the cosine similarity of that image's embedding (from InceptionNet) with any other image's embedding in the image pool is larger than 0.9/0.95.

**5. Manual Verification:** Following the automated cleaning, we manually inspect and verify the accuracy of the remaining images for each class to ensure they meet quality standards.

**6. Second-level De-duplication with Perceptual Hashing:** Post-verification, we use perceptual hashing [38] with a threshold of 10 bits to identify and remove duplicate images within each class, ensuring uniqueness across our dataset[5].

**7. Class Balancing:** Finally, we balance the dataset to ensure an equal representation of classes.

This process was followed for increased quality and reliability of our dataset for image recognition tasks.

---

[3]we use the image sourcing pipeline of C2C [96]

[4]We use the `fastdup` library for outlier removal: https://github.com/visual-layer/fastdup

[5]We use the `imagededup` library for de-duplication: https://github.com/idealo/imagededup

# I  Why and How Do We Use RAM++?

We detail why we use the RAM++ model [59] instead of CLIPScore [57] or open-vocabulary detection models [86]. Furthermore, we elaborate on how we selected the threshold hyperparameter used for identifying concepts in images.

## I.1  Why RAM++ and not CLIP or open-vocabulary detectors?

We provide some qualitative examples to illustrate why we chose RAM++. Our input images do not often involve complex scenes suitable for object detectors, but many fine-grained classes on which alongside CLIP, even powerful open-world detectors like OWL-v2 [86] have poor performance.

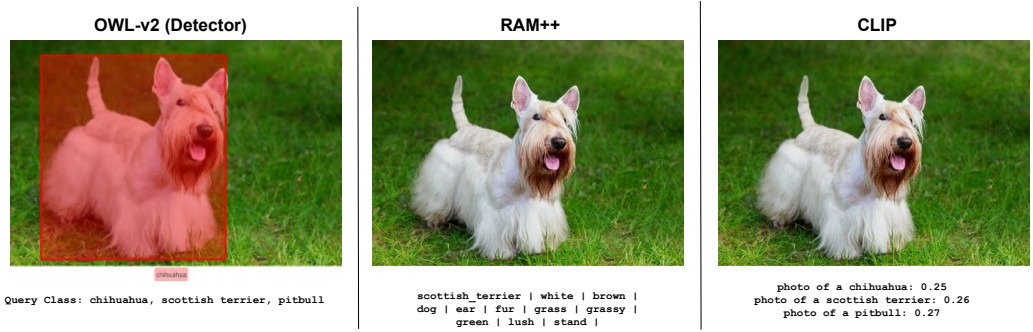

Figure 22: **Qualitative Results comparing OWL-v2, RAM++ and CLIP.** We show qualitative examples across three different models: OWL-v2, RAM++ and CLIP on fine-grained concepts.

## I.2  How: Optimal RAM++ threshold for calculating concept frequencies

We ablate the choice of the threshold we use for assigning concepts to images using the RAM++ model. For the given set of concepts, RAM++ provides a probability value (by taking a sigmoid over raw logits) for each concept's existence in a particular image. To tag an image as containing a particular concept, we have to set a threshold for deciding this assignment. We test over three thresholds: {0.5, 0.6, 0.7}, showcasing quantitative and qualitative results for all thresholds in Figs. 23 and 24.
We observe best frequency estimation results using the highest threshold of 0.7. This is due to the high precision afforded by this threshold, leading to us counting only the "most aligned images" per concept as hits. With lower thresholds (0.5, 0.6), we note that noisier images that do not align well with the concept can be counted as hits, leading to degraded precision and thereby poorer frequency estimation. To ensure that we do not incorrectly tag images with erroneous concepts, our primary objective is to optimize hit precision, even if it means occasionally missing out on tagging some images with correct concepts. Hence, we use 0.7 as the threshold for all our main results.

## I.3  GPT-4 Descriptions for each extracted concept

In addition to providing a list of concepts to the RAM++ model, we also provide a set of GPT-4 generated responses that describe each concept (please refer to Tab. 5 for examples). This ensures that we adequately cover synonyms of concepts and take into account concept hierarchies [84]. This further improves tagging precision by using visual descriptions to better identify concepts (this has been shown to enhance performance in previous works [97, 83]. We open-source these descriptions.

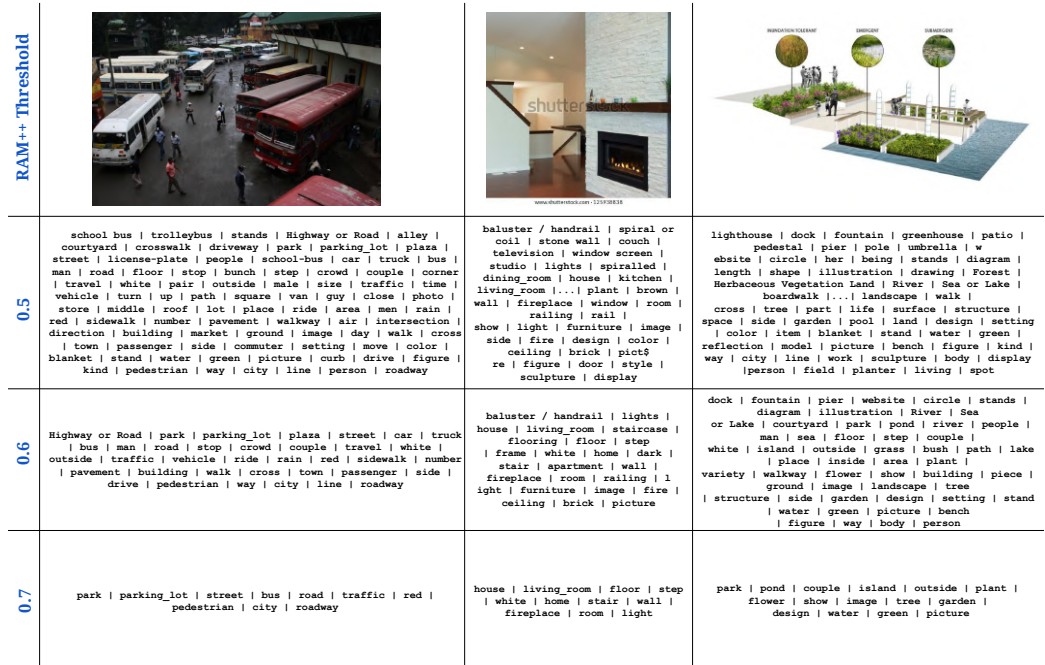

Figure 23: **Qualitative Results with different RAM++ thresholds.** We show qualitative examples across three different thresholds: {0.5, 0.6, 0.7} for estimating concept frequency using the RAM++ model. We note the significantly better concepts identified by the higher threshold (0.7) compared to the lower thresholds (0.5, 0.6). The images are sourced from the CC-3M dataset.

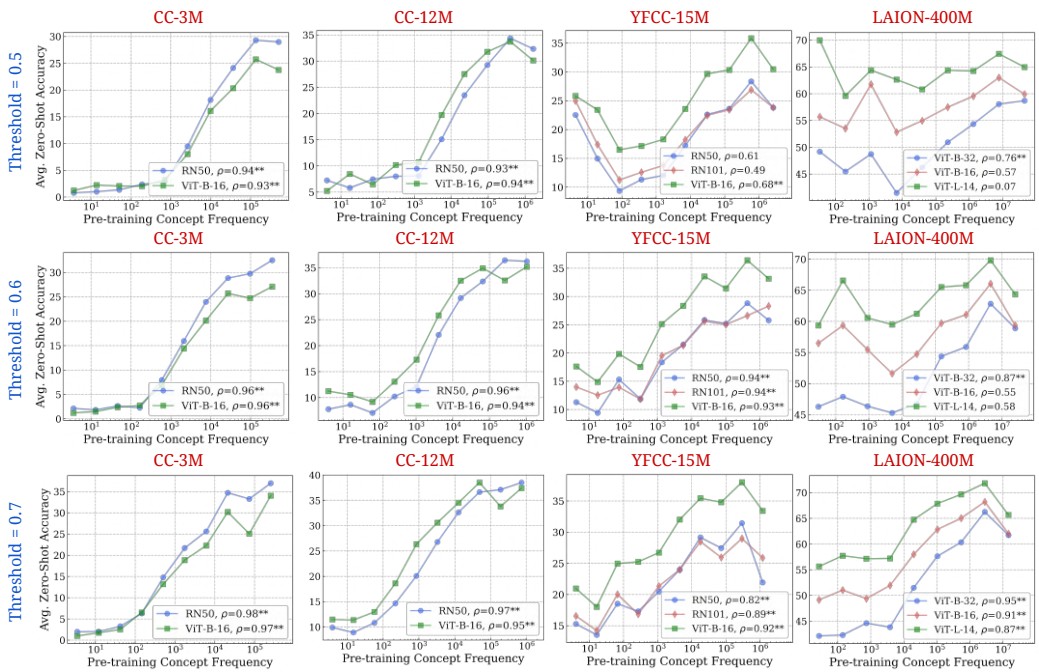

Figure 24: **Effect of different thresholds for determining concept frequency using RAM++.** We test three different thresholds: {0.5, 0.6, 0.7} for estimating concept frequency using the RAM++ model. Note that the lower the threshold, the lower precision are the tagged concepts since lower thresholds (0.5, 0.6) lead to noisier images being counted as hits, hence reducing the hit precision for determining frequency. Despite this added noise at lower thresholds, we note that all the correlations are significantly positive across all thresholds. This further signifies the robustness of our log-linear scaling trends inspite of our frequency estimates being noisy. ** indicates that the result is significant ($p < 0.05$ with two-tailed t-test.), and thus we show Pearson correlation ($\rho$) too.

Table 5: **Example GPT-4 Descriptions fed to RAM++ on a subset of downstream datasets and concepts**.

| Evaluation Dataset | Concept | GPT-4 Description |
|---|---|---|
| ImageNet [36] | Tench | A tench is a freshwater fish that typically has a greenish or brownish body with reddish fins, small scales, and a pair of barbels near its mouth. It can grow up to 70 cm long. |
| SUN397 [133] | Alley | An alley is a narrow passageway or lane between or behind buildings, which is often used for access or for parking. |
| UCF101 [116] | Blowing_Candles | A Blowing_Candles moment can be identified concisely as a moment or event where a person is blowing out the candles on a cake, typically at a birthday celebration. |
| Caltech101 [45] | Butterfly | A butterfly is a small, flying insect known for its colorful and symmetrical wings. It has a slender body, antennae and three pairs of legs. |
| CUB [131] | Least_Auklet | A Least Auklet is a small seabird with a black back and wings, white underparts, and a stubby orange bill. They also have white eye-rings and a small, rounded tail. |
| EuroSAT [56] | Pasture Land | Pasture land can be concisely identified as an open or cleared land covered with grass, clover, or the like, suitable for grazing by livestock, with little to no trees and is primarily used for agricultural purposes. |
| Flowers102 [90] | pink primrose | A pink primrose is a perennial flower featuring delicate, soft pink petals arranged around a yellow center, with bright green leaves at the base. |
| DTD [32] | bumpy | A bumpy object has an uneven or rough surface with lots of small raised areas or protuberances |
| Food101 [21] | churro | A churro is a long, thin, golden-brown pastry that is typically ridged and may be dusted with sugar. |
| FGVCAircraft [79] | 707320 | A 707320 is a model of the Boeing 707, which is a mid-size, long-range, narrow-body four-engine jet airliner. |
| Stanford-Cars [66] | 1993 Volvo 240 Sedan | The 1993 Volvo 240 Sedan can be identified by these features:1. Manufacturer: Volvo 2. Production Year: 1993 3. Model: 240 4. Body Style: 4-door sedan 5. Engine: 2.3L 4-cylinder 6. Transmission: 5-speed manual or 4-speed automatic |
| CIFAR100 [68] | bowl | A bowl is a round dish or container typically used to hold food, often deeper than a plate with a wide open top. |
| COCO-5K [75] | metro | A metro can be identified concisely as an urban railway system that operates within large cities, offering high-frequency services and utilizing multiple cars and stations. |

## J  Clarification regarding $0$-frequency points

In all our main plots, we explicitly exclude zero-frequency concepts from our evaluations following Kandpal et al. [62], since frequency estimation is potentially noisy, leading to low recall rates (also discussed in Appx. I). However, to verify if our log-linear trends still hold when including all the zero-frequency concepts, we re-plot all our main zero-shot classification results from Fig. 2 by including the ones which have zero-frequencies— Fig. 25 showcases these results. We find our main log-linear scaling trends are retained. To further corroborate this, we present average accuracies for concepts with frequency 0 and non-zero frequency bins in Tab. 6 below. We note that average performance for the 0-frequency concepts is significantly lower than other non-zero frequency concepts, especially when compared to very high-frequency concepts. This justifies our main claim that exponentially more data is needed per concept to improve performance linearly.

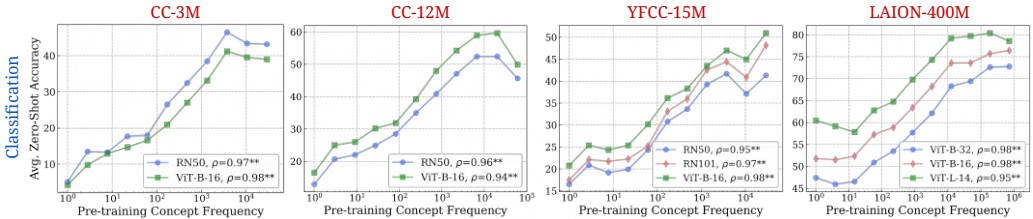

Figure 25: **Main Results with** $0$**-frequency concepts.** We re-plot all our main classification results from Fig. 2 by including concepts which have zero-frequencies. Note that the 0-frequency points are assimilated into the $10^0$ bin—in each plot, the $10^0$ bin (leftmost) consists of about $60 - 70\%$ 0-frequency concepts. We find that our main log-linear scaling trends are retained.

Table 6: **Performance per frequency bin.** Here, we explicitly report the average classification performance of models trained on different pretraining datasets, per frequency bin (*i.e.*, 0-frequency concepts only, concepts with frequencies in the range $1-10$, $10-100$ etc.). We note that average performance for the 0-frequency concepts is significantly lower than other non-zero frequency concepts, especially when compared to the performance of very high-frequency concepts.

| Dataset/Model | Freq=0 | Freq=1-10 | Freq=10-100 | Freq=100-1000 | Freq=1000-10000 |
|---|---|---|---|---|---|
| CC-3M/RN50 | 5.10 | 13.89 | 20.18 | 32.93 | 44.30 |
| CC-3M/ViT-B-16 | 4.27 | 11.98 | 17.21 | 27.48 | 39.24 |
| CC-12M/RN50 | 12.91 | 21.49 | 27.75 | 39.48 | 50.38 |
| CC-12M/ViT-B-16 | 16.48 | 25.59 | 32.07 | 45.65 | 57.06 |
| YFCC-15M/RN50 | 16.49 | 19.59 | 24.12 | 34.26 | 39.97 |
| YFCC-15M/RN101 | 17.43 | 22.06 | 25.72 | 36.77 | 43.14 |
| YFCC-15M/ViT-B-16 | 20.75 | 25.06 | 29.68 | 38.73 | 45.96 |
| LAION-400M/ViT-B-32 | 47.41 | 46.42 | 50.53 | 55.96 | 65.00 |
| LAION-400M/ViT-B-16 | 51.77 | 52.09 | 57.12 | 61.32 | 70.73 |
| LAION-400M/ViT-L-14 | 60.44 | 58.87 | 62.43 | 67.63 | 76.65 |

## K   Misalignment Degree Results and Human Verification

In Tab. 3 in the main paper, we quantified the *misalignment degree*, and showcased that a large number of image-text pairs in all pretraining datasets are misaligned. In Alg. 1, we describe the method used for quantifying this *misalignment degree* for each pretraining dataset. We also showcase some qualitative examples of a few image-text pairs from the CC-3M dataset that are identified as misaligned using our analysis in Fig. 26.

**Data:** Pretraining dataset $\mathcal{D} = \{(i_1, t_1), (i_2, t_2), \ldots, (i_N, t_N)\}$, Image Index $I_{\text{img}}$, Text Index $I_{\text{text}}$

**Result:** *mis_degree*

*mis_degree* $\leftarrow 0$
**for** $(i, t) \in \mathcal{D}$ **do**
    img_concepts $\leftarrow I_{\text{img}}[i]$ `// extract all concepts from this image`
    text_concepts $\leftarrow I_{\text{text}}[t]$ `// extract all concepts from this text caption`
    hits $\leftarrow$ `set_intersection`(img_concepts, text_concepts)
    **if** $len(hits) = 0$ **then**
        *mis_degree* $\leftarrow$ *mis_degree* $+ 1$
    **end**
    return *mis_degree*$/N$
**end**

**Algorithm 1:** Extracting *misalignment degree* from pretraining datasets

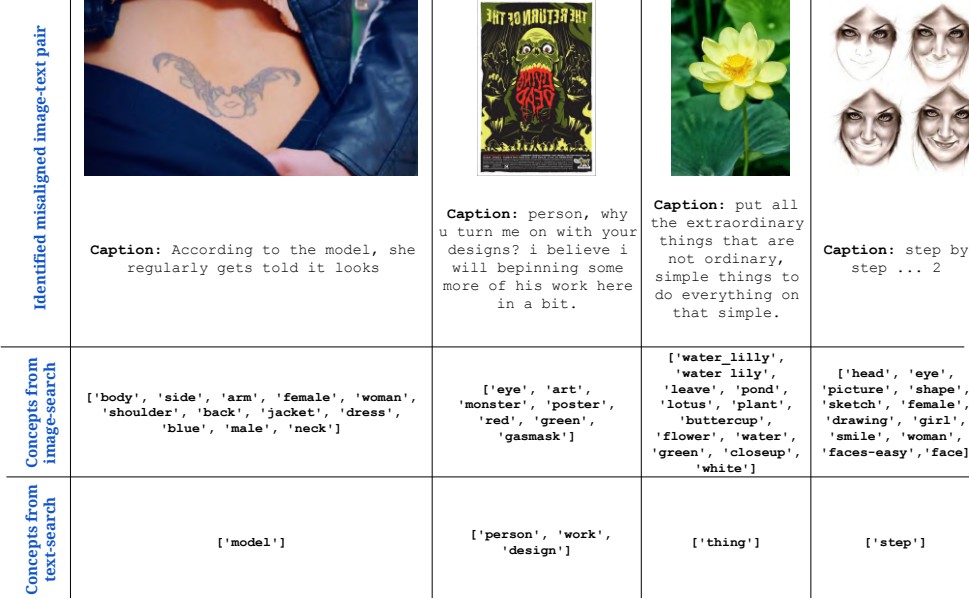

Figure 26: **Qualitative examples of misaligned image-text pairs identified.** We present 4 samples from the CC-3M pretraining dataset that are identified as misaligned by our analysis. Here, the text captions clearly do not entail the images, and hence do not provide a meaningful signal for learning.

**Human verification for misalignment results**. To verify the misalignment results from Tab. 3, we manually annotated 200 random image-text pairs from each dataset as aligned or misaligned. An image-text pair is misaligned if the text caption was irrelevant to the image. Previous work also found a similarly small random subset over large-scale web-datasets to be representative [78]. Our estimated misalignment results from Tab. 3 were in line with the human-verified results (see Tab. 7 below), corroborating our findings. Further, from our human-verification experiment, we found that the high misalignment degree in YFCC-15M is likely due to the lack of text quality filtering. YFCC-15M images are sourced directly from Flickr, where captions often provide high-level context rather than accurately describing the image content.

Table 7: **Human verification of mis-alignment results.**

| Dataset | Results from Tab. 3 | Human-verified results |
|---------|:-------------------:|:----------------------:|
| CC-3M | 16.81% | 18.00% |
| CC-12M | 17.25% | 14.50% |
| YFCC-15M | 36.48% | 40.50% |
| LAION-400M | 5.31% | 7.00% |

# L   Analysis of dips in high frequency concepts

We provide some intuitions on why there are some drops in the trend at high frequencies for the CC-3M and CC-12M classification plots in Fig. 2. We investigated which concepts occur at such high frequencies, specifically above $10^4$. From our analysis, we hypothesize two key reasons for these performance dips:

- ***Concept ambiguity:*** We observe many concepts that are homonyms / polysemous (same spelling but different meaning *i.e.*, can represent multiple concepts at once). Some examples are watch, bear, house, fly, bridge, cloud, park, face, bar, tower, wave, *etc*.

- ***Broad concepts:*** A concept with a broader scope of definition supersedes a narrower one (concept 'dog' vs the specific breeds of dogs seen in ImageNet ('yorkshire terrier', 'boston terrier', 'scottish terrier', 'golden retriever', etc)). These concepts are too coarse-grained and hence can be visually represented by a diverse set of images. Performance variance of these concepts can be quite high based on the specific set of images given for testing.

These ambiguities become more prevalent the more ubiquitous a concept is, which is directly tied to its frequency obtained from pretraining datasets. Some more examples for a deeper understanding of the diversity of concepts are: 'cucumber', 'mushroom', 'Granny Smith', 'camera', 'chair', 'cup', 'laptop', 'hammer', 'jeep', 'lab coat', 'lipstick', 'american-flag', 'bear', 'cake', 'diamond-ring', *etc*.

# M  Variance in performance per point in the zero-shot classification plots

We provide zero-shot classification plots for CC-3M, CC-12M, and LAION-400M in Fig. 27, including 95% confidence intervals for each point. This approach follows the standard practice from works like Miller et al. [85], Taori et al. [120]. Our plots show that the spread at higher frequencies is significantly larger than at moderate frequencies, following the analysis in Appx. L that higher frequency concepts are more ambiguous and polysemous. These results support the observed dips in accuracy at high-frequency points in the CC-3M and CC-12M plots in Fig. 2.

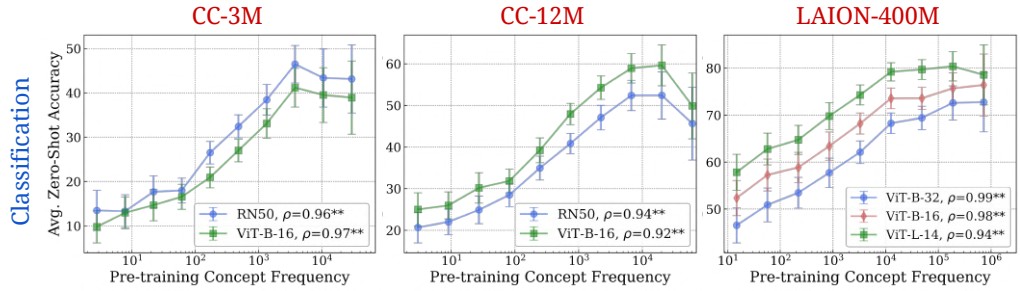

Figure 27: **Variance in performance per point in the zero-shot classification plots**

# N   T2I Models: Evaluation

We provide additional quantitative and qualitative results in this section for T2I models evaluated on the "*Let It Wag!*" dataset.

## N.1   Quantitative Results by Retrieval

We analyse how state-of-the-art T2I models perform on the long-tailed concepts comprising the "*Let It Wag!*" dataset. As detailed in Sec. 6, we generate 4 images for each concept using Stable Diffusion XL [95], Stable Diffusion v2 [104] and Dreamlike Photoreal [1].

**Prompting Strategy.** The prompting strategy (system role) used, adapted from Shahmohammadi et al. [113], was:

---

Follow my commands:
1. I wish to generate text prompts about a given subject which I will use for image generation using off-the-shelf text-to-image models such as Stable Diffusion and DALL-E 3.
2. Assume all the subjects are nouns.
3. Follow a similar style and length of prompts as coco-captions.
4. Keep prompts concise and avoid creating prompts longer than 40 words.
5. Structure all prompts by setting a scene with at least one subject and a concrete action term, followed by a comma, and then describing the scene. For instance,"a view of a forest from a window in a cosy room, leaves are falling from the trees."
Generate detailed prompts for the concepts in the order in which they are given. Your output should be just the prompts, starting with "1."

---

With this pool of generated images, we conduct a controlled experiment on the long-tailed concepts using nearest-neighbor retrieval as the evaluation metric by querying a generated image and retrieving the top-k results from a gallery of images taken from the "*Let It Wag!*" dataset. The overall pipeline is as follows:

**Setup.** We define the query and gallery set for head and tail concepts. For tail concepts, we sample the 25 concepts with the lowest frequency from the "*Let It Wag!*" dataset. For head concepts, we sample the 25 most frequent concepts for comparison. We use the same prompting strategy with the selected 25 concepts across all 3 T2I models. To create the gallery set, we randomly sample 100 images for each of these concepts. We use DINOv2 [91] ViT-S/14 as the feature extractor.

**Results.** In Table 8, we provide the Cumulative Matching Characteristic (CMC@k) results for all 3 T2I models used in our experiment. CMC@k was chosen as we are interested in measuring the performance delta between head and tail concepts for successful retrievals within the top-k retrieved real images for a given generated image. We observe a large performance gap between *Head* and *Tail* concepts, providing a quantitative evaluation of generation performance of T2I models.

Table 8: **Generated-real retrieval scores.** We compare retrieval results of DINOv2 ViT-S/14 when using generated images as query images. We report $\Delta$ CMC@k results where k={1,2,5} between head and tail concepts.

| Model | $\Delta$CMC | | |
|---|---|---|---|
| | k=1 | k=2 | k=5 |
| Stable Diffusion XL | 13.0 | 16.0 | 16.8 |
| Stable Diffusion v2 | 11.0 | 10.0 | 10.4 |
| Dreamlike Photoreal | 8.0 | 9.0 | 9.4 |

## N.2   Qualitative Results

In Fig. 7 of the main text, we provide an initial insight into the qualitative performance of T2I models on *"Let It Wag!"* concepts. For ease of comprehension and comparison, we segregate concepts

into 4 clusters: `Aircraft` (Fig. 28), `Activity` (Fig. 29), `Animal` (Fig. 30) and others (Fig. 31). **Please note that we compress the aforementioned images to a lower quality due to the file size limitation of our submission. We will replace them with the original, high quality image files for the final version.**

**Results.** Fig. 28 shows T2I models having difficulty in representing an aircraft in its full form in a majority of cases in addition to misrepresenting the specific model in the generated images. Fig. 29 showcases the difficulty T2I models face when representing actions or activities from prompts. Fig. 30 exemplifies the same inability of T2I models to accurately represent animal species. Finally, the remainder of the query set is shown in Fig. 31 and includes the inability to classify and subsequently generate certain species of flowers and objects.

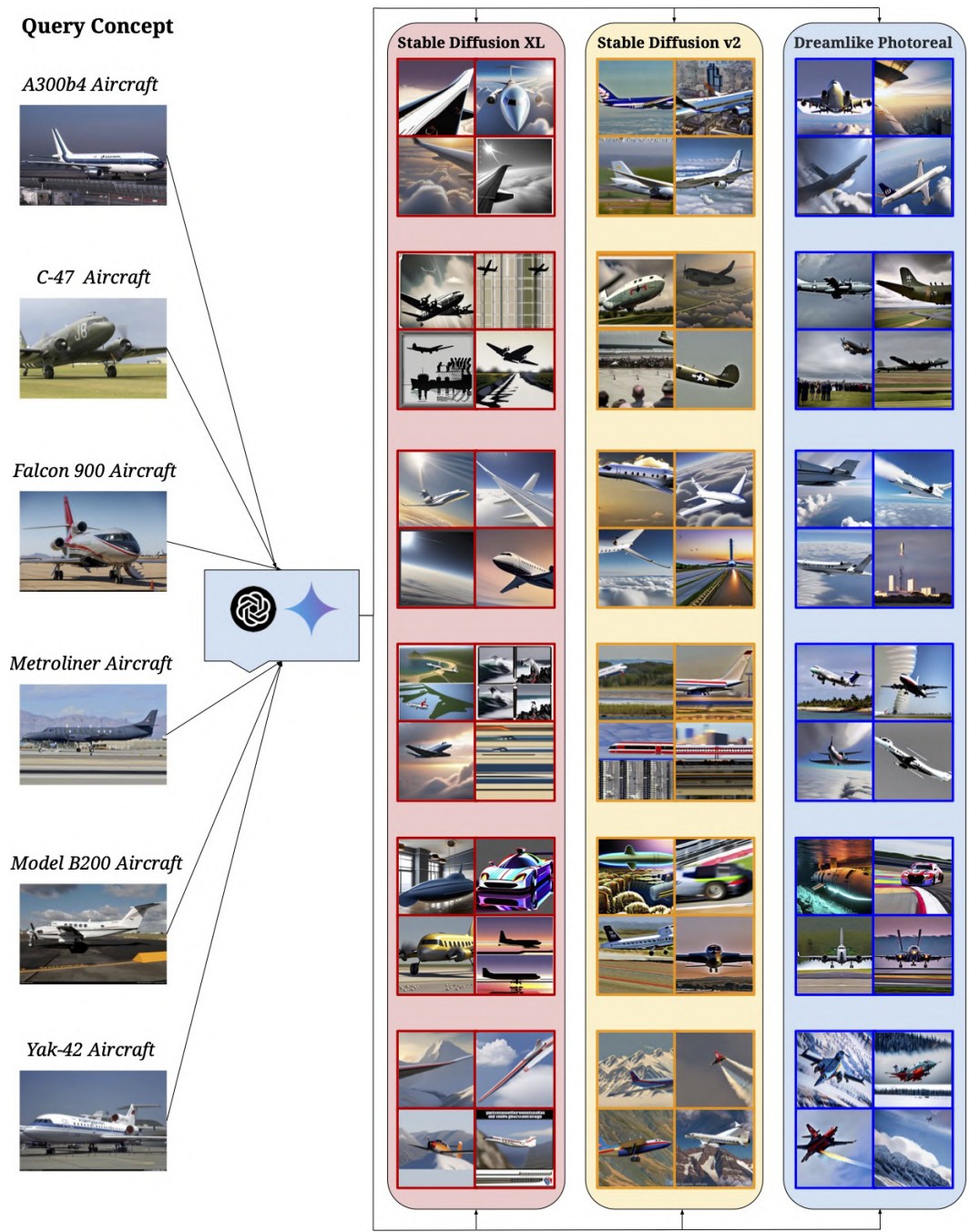

Figure 28: **Qualitative results on the** `Aircraft` **cluster**.

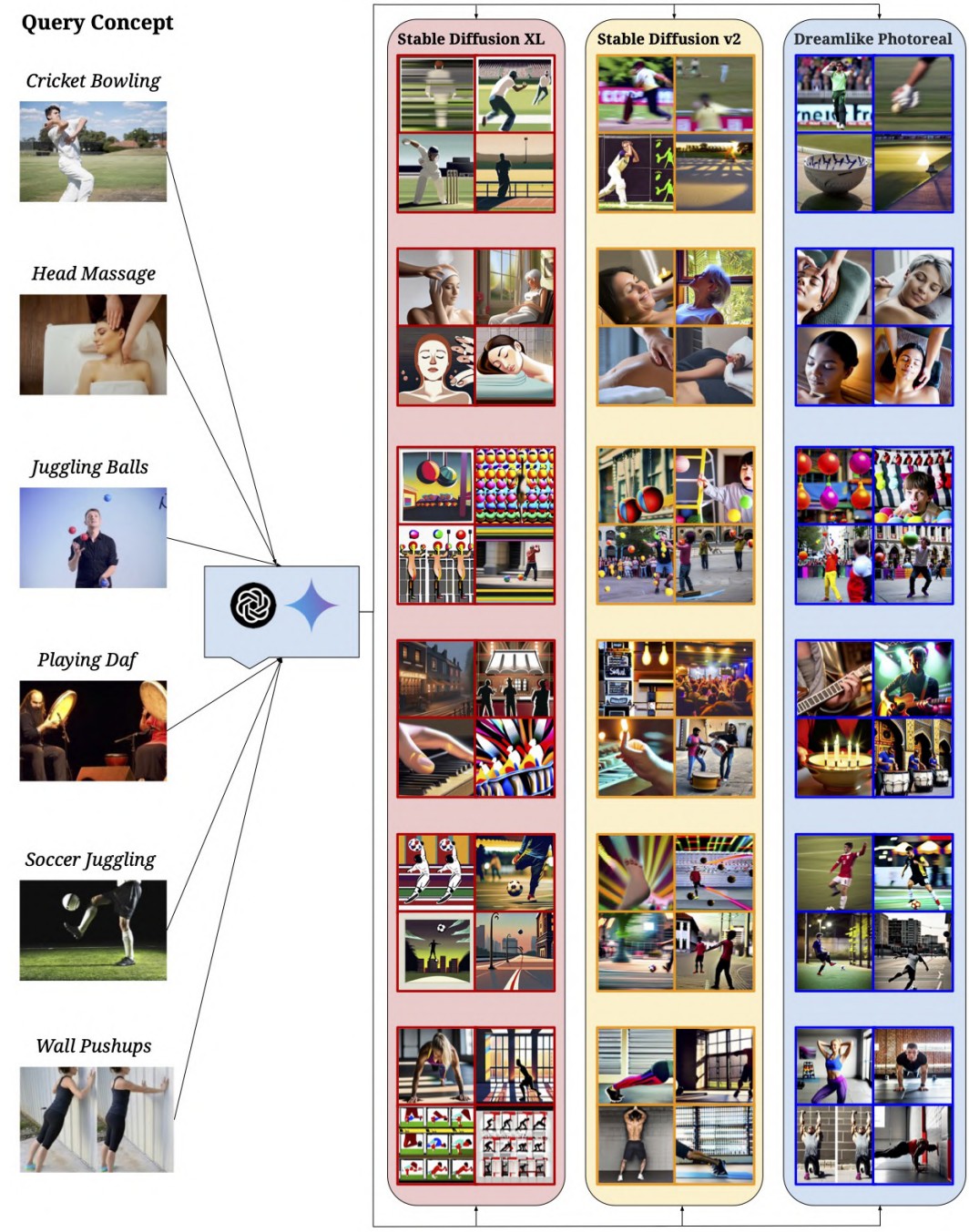

Figure 29: **Qualitative results on the** `Activity` **cluster.**

**Query Concept**

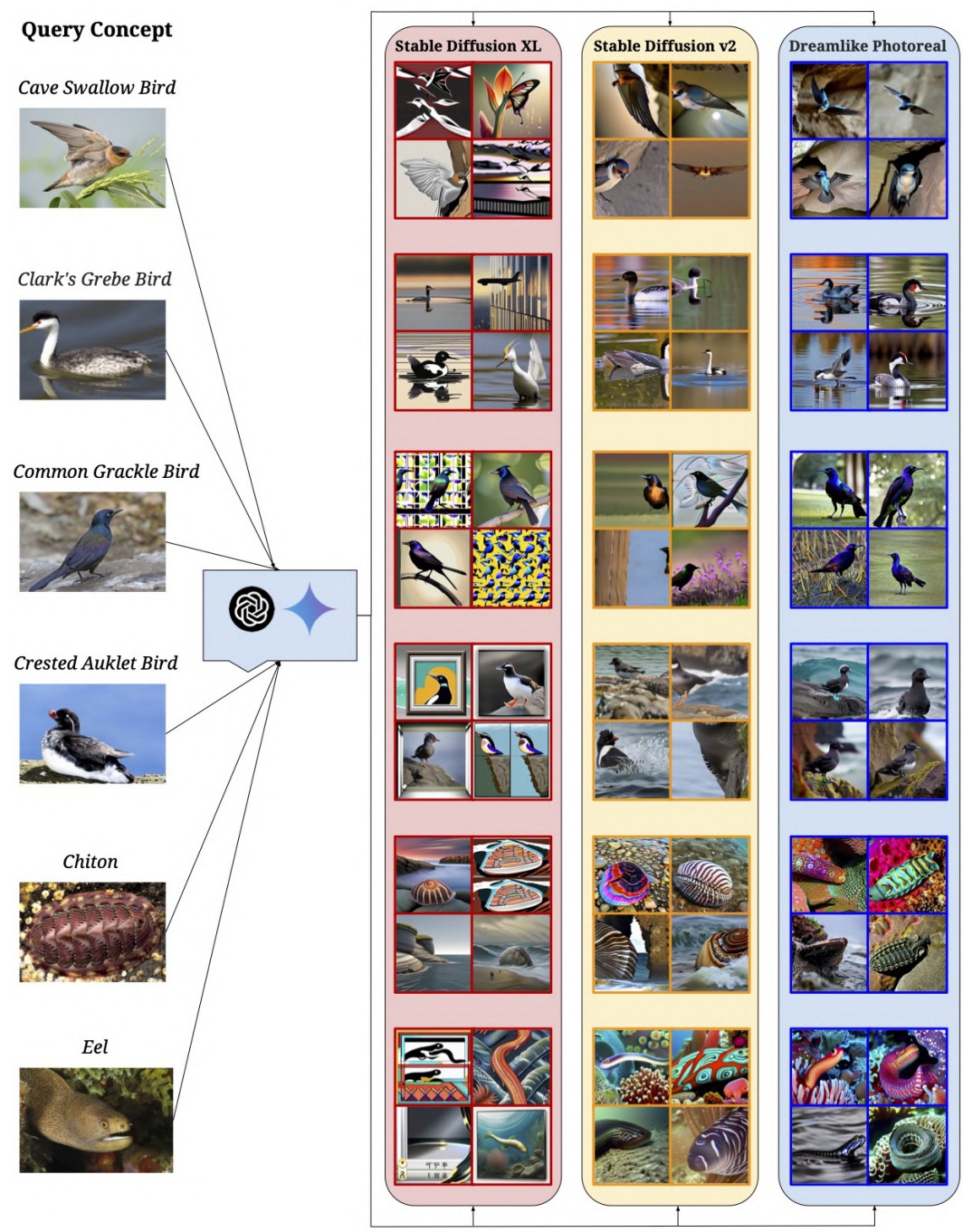

*Cave Swallow Bird*

*Clark's Grebe Bird*

*Common Grackle Bird*

*Crested Auklet Bird*

*Chiton*

*Eel*

Stable Diffusion XL

Stable Diffusion v2

Dreamlike Photoreal

Figure 30: **Qualitative results on the** `Animal` **cluster.**

**Query Concept**

*Bishop of Llandaff Flower*

*Cautleya Spicata Flower*

*Thorn Apple Flower*

*Amphibious Vehicle*

*Guillotine*

*Hairspray*

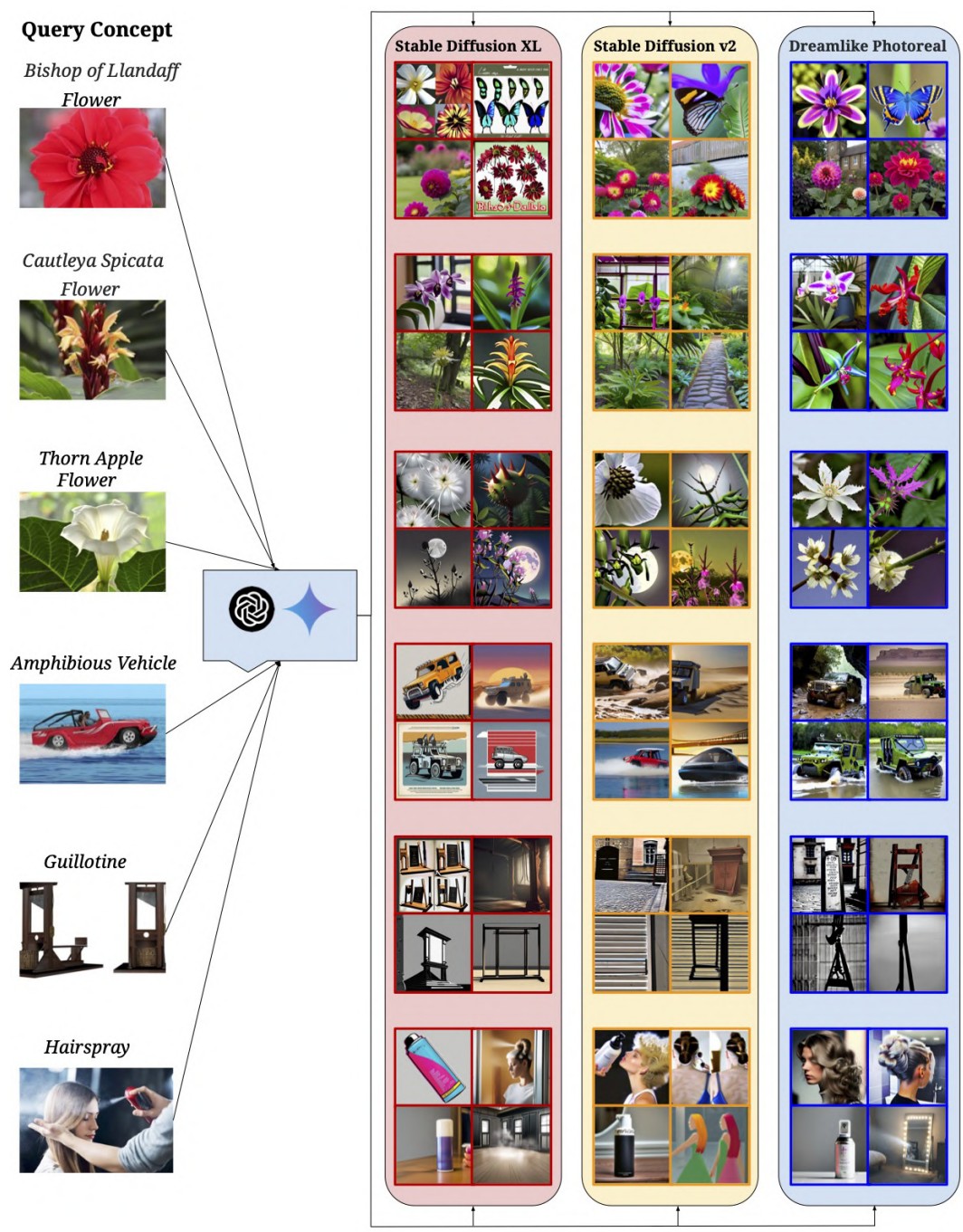

Figure 31: **Qualitative results for other selected failure cases.**

# O   Classification Results: *Let It Wag!*

Here, we present the raw accuracy values of the 40 tested models on both *Let It Wag!* and ImageNet in Tab. 9. For reference, we also report the datasets these models were trained on and the number of parameters for each model. We see clear drops in performance compared to ImageNet, across model sizes, architectures and pretraining datasets.

Table 9: Full results dump on *Let It Wag!* and ImageNet.

| Pretraining Dataset | Model | Num. Parameters (in millions) | ImageNet Acc. | *Let It Wag!* Acc. |
|---|---|---|---|---|
| CC-3M [115] | RN50 | 102.01 | 20.09 | 3.74 |
| | ViT-B-16 | 149.62 | 17.10 | 3.01 |
| CC-12M [28] | RN50 | 102.01 | 33.14 | 8.92 |
| | ViT-B-16 | 149.62 | 37.39 | 11.49 |
| YFCC-15M [123] | RN50 | 102.01 | 31.88 | 13.15 |
| | RN101 | 119.69 | 34.04 | 15.19 |
| | ViT-B-16 | 149.62 | 37.88 | 19.25 |
| OpenAI-WIT [98] | RN50 | 102.01 | 59.82 | 31.93 |
| | RN101 | 119.69 | 62.28 | 31.88 |
| | ViT-B-32 | 151.28 | 63.32 | 33.52 |
| | ViT-B-16 | 149.62 | 68.34 | 37.85 |
| | ViT-L-14 | 427.62 | 75.54 | 45.31 |
| WebLI [30] | ViT-B-16 | 203.79 | 78.49 | 54.63 |
| | ViT-L-16 | 652.15 | 82.07 | 61.50 |
| | SO400M | 877.36 | 83.44 | 67.32 |
| DataComp [47] | ViT-B-32 | 151.28 | 69.18 | 46.90 |
| | ViT-B-16 | 149.62 | 73.48 | 52.89 |
| | ViT-L-14 | 427.62 | 79.21 | 63.04 |
| DataComp-DFN [44] | ViT-B-16 | 149.62 | 76.24 | 56.59 |
| | ViT-H-14 | 986.11 | 83.44 | 71.91 |
| CommonPool [47] | ViT-B-32 | 151.28 | 23.04 | 7.73 |
| | ViT-B-16 | 149.62 | 57.77 | 20.97 |
| | ViT-L-14 | 427.62 | 76.37 | 46.96 |
| LAION-400M [110] | ViT-B-32 | 151.28 | 60.23 | 32.88 |
| | ViT-B-16 | 149.62 | 67.02 | 39.13 |
| | ViT-L-14 | 427.62 | 72.74 | 46.59 |
| LAION-2B [111] | ViT-B-32 | 151.28 | 66.55 | 41.79 |
| | ViT-B-16 | 149.62 | 70.22 | 44.21 |
| | ViT-L-14 | 427.62 | 75.25 | 51.03 |
| | ViT-H-14 | 986.11 | 77.92 | 58.98 |
| | ViT-g-14 | 1366.68 | 78.46 | 59.01 |
| | ViT-bigG-14 | 2539.57 | 80.09 | 63.54 |
| MetaCLIP-400M [135] | ViT-B-32 | 151.28 | 65.58 | 40.50 |
| | ViT-B-16 | 149.62 | 70.80 | 46.50 |
| | ViT-L-14 | 427.62 | 76.20 | 52.78 |
| MetaCLIP-FullCC [135] | ViT-B-32 | 151.28 | 67.66 | 43.84 |
| | ViT-B-16 | 149.62 | 72.12 | 49.32 |
| | ViT-L-14 | 427.62 | 79.17 | 57.48 |
| | ViT-H-14 | 986.11 | 80.51 | 62.59 |
| SynthCI-30M [52] | ViT-B-16 | 149.62 | 30.67 | 9.15 |

## P   Compute and Storage Resources

We run all our RAM++ image index construction and search experiments using NVIDIA A-100-80GB, 2080-TI and A-100-40GB GPU nodes. For the text index construction and search experiments, we use a CPU server with a 48-core Intel Xeon Platinum 8268 CPU and 392GB of RAM. We document the precise storage and compute costs for all our experiments, pertaining to each pretraining dataset used, in Tab. 10.

Table 10: **Compute and Storage Resources Utilized**. We report the total disk space required for storing all pretraining datasets along with the number of shards stored. Further, we also report the exact wall-clock runtimes (WCT) for running the RAM++ image tagging scripts and the text-index construction across all downstream datasets, on a single GPU/CPU node.

| Pretraining Dataset | Disk-Space | Number of Shards | RAM++ WCT | Text-Index Const. WCT |
|---|---|---|---|---|
| CC-3M | 243GB | 332 | 16h | 54h |
| CC-12M | 1.2TB | 1100 | 55h | 216h |
| YFCC-15M | 1.1TB | 1500 | 75h | 270h |
| LAION-400M | 9.4TB | 41408 | 2070h | 7200h |
| LAION-A | 5.4TB | 16110 | 805h | 2700h |
| SynthCI30M | 527GB | 3040 | 101h | 540h |

# Q Licenses and Attributions

In this section, we credit the owners of all assets (datasets and models) used in our experiments and also provide the license of each of these assets. Please refer to Tabs. 11 and 12.

Additionally, we also provide attributions for each icon used in Fig. 1 as detailed below. Each icon is free to use for commercial and non-commercial applications with attribution.

- Neural network icons created by Freepik - Flaticon
- Folder icons created by Freepik - Flaticon
- Retrieval icons created by Prosymbols Premium - Flaticon
- Database icons created by Freepik - Flaticon
- Paintbrush icons created by nawicon - Flaticon

Table 11: **Licenses for all pretraining and downstream datasets used in this work**.

| Dataset | Source | License |
|---|---|---|
| CC-3M | [115] | Custom License |
| CC-12M | [28] | Custom License |
| YFCC-15M | [123] | Creative-Commons |
| LAION-400M | [110] | CC-BY-4.0 |
| LAION-A | [111] | CC-BY-4.0 |
| SynthCI-30M | [52] | CC-BY-NC-4.0 |
| ImageNet | [36] | Custom Non-Commercial |
| SUN397 | [133] | Unknown |
| UCF101 | [116] | CC-0 Public Domain |
| Caltech101 | [45] | CC-BY-4.0 |
| EuroSAT | [56] | CC-BY-4.0 |
| CUB | [131] | CC-0 Public Domain |
| Caltech256 | [50] | CC-BY-4.0 |
| Flowers102 | [90] | Unknown |
| DTD | [32] | Unknown |
| Birdsnap | [16] | Unknown |
| Food101 | [21] | Unknown |
| Stanford-Cars | [66] | Unknown |
| FGVCAircraft | [79] | Custom Non-Commercial |
| Oxford-Pets | [93] | CC BY-NC-SA-4.0 |
| Country211 | [98] | Creative-Commons |
| CIFAR-10,CIFAR-100 | [68] | Unknown |
| Flickr-1K | [138] | CC-0 Public Domain |
| COCO-5K,COCO-Base | [75] | CC-BY-4.0 Legal-Code |
| CUB200 | [131] | CC-0 Public Domain |
| Daily-DALLE | [34] | Apache-2.0 |
| Detection | [31] | MIT |
| Parti-Prompts | [140] | Apache-2.0 |
| DrawBench | [106] | Unknown |
| Relational Understanding | [33] | Unknown |
| Winoground | [124] | Custom License |

Table 12: **Licenses for all models used in this work**.

| Model | Source | License |
|---|---|---|
| ViT-B-16, ViT-B-32, ViT-L-14 | [37] | Apache-2.0 license |
| ResNet50, ResNet101 | [54] | MIT License |
| M-Vader | [15] | Unknown |
| DeepFloyd-IF-M, DeepFloyd-IF-L, DeepFloyd-IF-XL | [9] | DeepFloyd IF License Agreement |
| GigaGAN | [63] | Unknown |
| DALL·E Mini,DALL·E Mega | [35] | Apache-2.0 license |
| Promptist+SD-v1.4 | [53] | MIT |
| Dreamlike-Diffusion-v1.0 | [2] | Unknown |
| Dreamlike Photoreal v2.0 | [3] | Unknown |
| OpenJourney-v1 | [4] | CreativeML OpenRAIL License |
| OpenJourney-v2 | [5] | CreativeML OpenRAIL License |
| SD-Safe-Max,SD-Safe-Medium,SD-Safe-Strong,SD-Safe-Weak,SD-v1.4,SD-v1.5, SD-v2-Base,SD-v2-1-Base | [104] | CreativeML OpenRAIL License |
| Vintedois-Diffusion-v0.1 | [7] | CreativeML OpenRAIL License |
| minDALL.E | [105] | Apache-2.0 license |
| Lexica-SD-v1.5 | [1] | CreativeML OpenRAIL License |
| Redshift-Diffusion | [6] | CreativeML OpenRAIL License |

# R  Limitations, Open Questions and Future Directions

We highlight a few limitations and open questions of our work, leading to some possible exciting avenues for future research.

**Understanding Image-Text Misalignments.** One can explore the origins of misalignments between images and texts, such as the limitations of exact matching for concept identification in captions, inaccuracies from the RAM++ tagging model, or captions that are either too noisy or irrelevant. A few potential mitigating strategies are to explicitly recaption the images [29, 89] or to utilize the grounded concepts from the images as aditional feedback signal.

**Investigating Compositional Generalization.** In our work, we only analyse concepts in isolation, and do not take into account the combination of concepts. "Zero-shot generalization" often refers to models' ability for compositional generalization (understanding new combinations of concepts not previously encountered). This is distinct from traditional zero-shot learning and presents an intriguing, yet unresolved challenge: analyzing compositional generalization from a data-centric perspective.

**Methods for Bridging the Generalization Gap.** Addressing the challenges posed by the long-tail distribution involves improving model generalization to overcome the limited improvement from pretraining we found in our study. Retrieval mechanisms can compensate for the inherent generalization shortcomings of pretrained models, providing a viable path to mitigating the effects of long-tailed pretraining data distributions.

**Towards a Theoretical Model for the Log-Linear Scaling Trends.** Our experiments comprehensively showcase the log-linear scaling trend of model performance with pretraining concept frequency empirically, across several diverse pretraining datasets and models. However, our analysis lacks a detailed theoretical framework explaining why such a trend exists. Building such a framework can help get better intuitions about the underlying mechanics of data dependence in multimodal models, which could be crucial for developing more efficient training strategies or algorithms.

**On the Interaction of Model Scale and Concept Frequency.** An important aspect of the current recipe for building robust foundation models is model scale. Despite investigating models across different scales, a key open question is what the effect of model scaling would be on the slope of the log-linear fit in our plots. Precisely studying the rate of change of the slope across model scales would enable making stronger claims on the optimal capacity-data-frequency tradeoffs.

**Potential Mitigating Solutions.** While our paper does not propose specific solutions, we believe its primary contribution is in thoroughly highlighting the issues with current pretraining strategies for multimodal models across various datasets, pretraining methods, architectures, training objectives, and tasks. Additionally, by releasing the *"Let it Wag!"* testbed, we provide a straightforward test set for future research to build upon, aiming to improve the generalization of multimodal models to long-tail scenarios. However, we suggest a few potential methods that could be explored to enhance multimodal long-tail:

- *Retrieval Augmentation:* Enhancing generalization to long-tail concepts can be achieved by utilizing the "world-knowledge" of LLMs to provide detailed descriptions for these concepts. This approach transforms the task from simply recognizing long-tail concepts by name to recognizing them by both names and descriptions.

- *Curriculum Learning:* Our tested models used random IID sampling during training. However, research into better sequencing of data samples could potentially improve model generalization to long-tail concepts by inducing more transferable feature representations in VLMs.

- *Synthetic Data:* Addressing the issue of long-tail concepts in web-sourced datasets may not be feasible by merely increasing data samples. There will likely always be low-data density regions in the pretraining data distribution. Using synthetic data, either through procedurally generated samples or text-to-image models, could be a viable mitigation strategy.

We hope these suggestions provide valuable directions for future research and contribute to the development of multimodal models capable of better generalization.

# S  Broader Impacts

Our work uses large-scale image-text pretraining datasets and models. The broad societal implications of both of these artifacts have been comprehensively discussed in prior work [98, 19, 20]. By extensively studying the composition of these large-scale datasets via principled methods, our work tries to gain a better understanding of their composition. A key result from our work that has serious potential implications for the broader society is the poor performance of multimodal models on the long-tail. From Tab. 4 and Fig. 5, it is clear that web-sourced datasets all exhibit the same long-tailed biases. This suggests that current models will predictably underperform on digitally marginalized communities and societies that are underrepresented on the web. Our results call for improved algorithms for training such multimodal models, such that they are more inclusive and performant on the long-tail. We also publicly release all of our data artifacts. Since the multimodal datasets we analyze in our work are extremely biased and can contain hateful, harmful and toxic content [20], our publicly released data artifacts potentially reflect these biases too. However, we hope that, by facilitating analysis of such large-scale datasets via our artifacts, future research efforts focus on gaining a better understanding of how to make these datasets fairer and more inclusive.

