# OpenReview forum: "No "Zero-Shot" Without Exponential Data: Pretraining Concept Frequency Determines Multimodal Model Performance"
_NeurIPS.cc/2024/Conference — NeurIPS 2024 poster_

### Official Review · Reviewer_KUaM · 2024-07-03

**Soundness:** 2
**Presentation:** 2
**Contribution:** 2
**Rating:** 6
**Confidence:** 3

**Summary:**

This paper investigates the "zero-shot" performance of multimodal models like CLIP and Stable-Diffusion. By analyzing 34 models and 5 pretraining datasets, the study finds that these models require exponentially increasing amounts of data to achieve linear improvements in performance. Additionally, the study suggests that the key to effective "zero-shot" generalization with large-scale training data and compute paradigms remains elusive.

**Strengths:**

- The authors conducted a comparative analysis involving two main factors: (1) the performance of models across various downstream tasks, and (2) the frequency of test concepts within their pretraining datasets.

- This paper is easy to read.

- There is ample work involved in this study.

**Weaknesses:**

-  Please unify the description of the table, above or below the table.

- This work does not point out correlations or differences with prior work, preventing me from assessing the novelty of this work.

-  Qualitative visualizations and relevant theoretical analysis are lacking, although the perspective of the work is easy to understand.

**Questions:**

- Does this work conflict with research into improving the generalization of multimodal models? I think the author should discuss it.

- Explanation of Figures 24-27 is insufficient.

- What is the motivation behind this work? What implications does it have for future research?

- To validate the points made in this paper, should other tasks beyond classification and retrieval be evaluated?

**Limitations:**

- See Weaknesses and Questions.

---

> ### Author Rebuttal · Authors · 2024-08-07
>
> > **W1: unify table descriptions**
>
> Thank you for pointing out this discrepancy, we apologise for the oversight and have fixed this in the paper.
>
> > **W2: connections to existing work**
>
> Thank you for this question. We have included a brief related work section in the main paper in sec 7, describing the differences of our work to the other related work in literature. Further, we discussed our paper’s position in the literature in the last paragraph of the introduction under, "Situating our Contributions in Broader Literature." Lastly, we note that we have included a larger related works section in appx E. We hope this addresses the concern.
>
> > **W3: qualitative visualizations/theoretical analysis lacking**
>
> - **Regarding qualitative visualizations:** While we share T2I generations for the long-tailed "Let It Wag!" concepts in figs 7, 24-27 as well as misalignment in fig 23, we are happy to provide additional figures visualizing concepts according to frequency in the revised paper.
> - **Regarding relevant theoretical analysis:** While we mention that theoretical analysis is lacking explicitly in our limitations (see appx N), our work is an example of empirical theory [[1](https://arxiv.org/abs/2001.08361),[2](https://openreview.net/forum?id=AssIuHnmHX)], where we attempt to quantitatively describe real-world behavior, thereby yielding conjectures, which, if proven, yields theory that holds in practice. Such approaches are standard in physics, and has yielded characterizations that have become prescriptive for practitioners ([Hoffmann et al](https://openreview.net/forum?id=iBBcRUlOAPR)). In our case, we have shown that typical power law (log-log) relationshisp between training set size and performance do not hold when downstream performance is related to concept frequency within training sets, where we instead observe log-linear trends, thereby characterizing the detrimental effects of the long tail. We will expand upon this in the revised version of the paper.
>
> > **Q1: conflicting with works that improve multimodal model generalization?**
>
> Thank you for this insightful question.
> We believe our conclusions should hold regardless of the model architecture and the training objective for any VLM. However, to test this, we investigated two methods that have been empirically shown to improve generalization capabilities of CLIP models: [CyCLIP](https://arxiv.org/abs/2205.14459) and [SLIP](https://arxiv.org/abs/2112.12750). We use 4 different models, each trained with either CyCLIP/SLIP on three different datasets---we then plot our main log-linear scaling results similar to figs 2, 3 for CyCLIP and SLIP models—these plots are in the uploaded rebuttal PDF at the bottom. We observe for both SLIP and CyCLIP models, **the log-linear scaling trends hold strong, with high Pearson correlation coefficients**, further signifying the robustness of our main results. Hence, we emphasize that **our main conclusions hold true even when considering multimodal models that explicitly introduce new training objectives with the aim of improving model generalization**.
>
> > **Q2: insufficient explanation of figs 24-27**
>
> We apoloigise for the lack of clarity. The motivation behind figs 24-27 was to provide qualitative insights into failure of T2I models on long-tailed concepts, providing a more detailed analysis of fig 7 of the paper. We used three T2I models to generate images from prompts generated for each concept by Gemini/GPT4. For a more holistic understanding of model failures on long-tailed concepts, we group concepts into a broader semantic category: aircraft (fig 24), activity (fig 25), animal (fig 26) and misc (fig 27), to showcase the broad spectrum of concepts captured and how T2I models fail in each case. This has been detailed in sec 6 of main paper and appx. J.2 but we will expand the section further with clarifying details.
>
> > **Q3: motivation and implications of work?**
>
> Thanks for this question. While we provide motivation in our introduction and have also included broader impacts section (appx O) to highlight the implications of our work for future research in multimodal models, we will make sure to clarify these sections further.
> - **Motivation:** While multimodal models demonstrate impressive "zero-shot" performance, knowing whether this is a result of an overlap between downstream tasks and the pretraining dataset not only calibrates expectations for how these models will perform in-the-wild, but also clarifies if the aforementioned performance could be achieved more efficiently by leveraging improved dataset priors, ie downstream task-aware curation.
> - **Implications:** Given the observed sample inefficient log-linear scaling trend suggests that models indeed are reliant on downstream task concepts being exponentially frequent in their pretraining dataset, this implies that reliable performance when encountered with rare concepts remains an outstanding question for current multimodal foundation models. Based on this analysis, we contribute the "Let-It-Wag!" testbed by aggregating rare concepts, to further future research in this direction.
>
> > **Q4: tasks beyond classification/retrieval**
>
> Thank you for the question—we agree, it is important to consider tasks outside classification/retrieval. Hence, we have studied the performance of 24 T2I models on the task of text-based image-generation and observed the same log-linear performance trend as that of classification and retrieval. Please refer to fig 3 (main paper) and fig 11-15 (appendix) for results of the log-linear claim for all metrics across all 24 text-to-image models. Additionally, we provide a human evaluation on the text-to-image results as well, please see appendix C and fig 16, 17. Finally, we also highlight the difference in performance between head and tail concepts using a quantitative nearest-neighbour retrieval experiment—please refer appendix J. We hope this sufficiently addresses the reviewer’s concern.

---

### Official Review · Reviewer_WCHb · 2024-07-10

**Soundness:** 3
**Presentation:** 4
**Contribution:** 4
**Rating:** 7
**Confidence:** 5

**Summary:**

The authors evaluate CLIP's zero-shot performance and its correlation to the pretraining data. Their experiments, based on the open LAION datasets, show that the limitations of Vision-Language Models (VLMs) in downstream tasks such as image generation and zero-shot recognition are linked to the frequency of concepts in the data.

VLMs like CLIP are pivotal for downstream multimodal applications, and understanding their biases is crucial for developing more robust systems. While the paper demonstrates the limitations of these foundational models, it currently focuses more on breadth rather than presenting in-depth insights.

I would be willing to increase my score if the authors could address my questions and fix some of the references.

[1] Parashar, Shubham, et al. "The Neglected Tails in Vision-Language Models." Proceedings of the IEEE/CVF Conference on Computer Vision and Pattern Recognition. 2024.

[2] Schuhmann, Christoph, et al. "Laion-5b: An open large-scale dataset for training next generation image-text models." Advances in Neural Information Processing Systems 35 (2022): 25278-25294.

**Strengths:**

1. The paper is well-written and motivated, highlighting the importance of understanding the limitations of Vision-Language models like CLIP.

2. The paper uniquely demonstrates that the generation capabilities of T2I models are correlated with the frequency of a given concept, which is novel and insightful.

3. The authors also introduce a new dataset that aggregates tailed concepts from Vision-Language Model pretraining datasets.

**Weaknesses:**

1. I feel the authors are somewhat aggressive in their claim of a 'constant linear relationship' as stated in Fig 2's caption. There are noticeable dips in accuracy that are not well explained. Given that the authors can identify false positives using their frequency measurement pipeline and are confident about the frequency of a given concept, a true linear correlation should not show these dips. This observation is not adequately addressed by the authors.

2. The authors' claim that their work is the first to establish that pretraining datasets of VLMs are long-tailed seems overzealous. Prior work has demonstrated this fact [1], and the correlation between zero-shot accuracy and pretraining frequency has been established. Additionally, [1] analyzes both LAION-400M and LAION-2B, a superset of LAION-400M [2], whereas the biggest dataset evaluated in this work is LAION-400M. Furthermore, [1] has been incorrectly cited in the paper.

3. The authors present a challenging new benchmark but provide very limited information on how these concepts were collected. There could be multiple ways to sample 290 classes, which could lead to worse performance for CLIP compared to popular benchmarks like ImageNet.

4. Additionally, the authors could have provided some high-level information about which concepts are relatively less present, such as snakes, birds, airplanes, etc. This information could offer insights toward improving CLIP.

**Questions:**

1. Can the authors provide insights about the classes used to create the dataset? Specifically, what was (a) the most frequent concept, (b) the least frequent concept, and (c) the average frequency of a concept? This would help understand the structure of the dataset, and how tailed the dataset is.

2. Can the authors present the standard deviation of each frequency bin? It would be interesting to see the variance as the frequency increases, as this may help explain the dips. CLIP might show less variance for relatively frequent concepts.

**Limitations:**

1. While the authors highlight a significant problem, they do not propose any solutions to mitigate it. Offering potential solutions could have strengthened the paper.

2. The paper covers a broad range of topics, but it would benefit from a stronger focus on the generation aspect and dataset. Previous literature [1] has indicated that zero-shot recognition of CLIP models depends on the distribution of concepts in the pretraining data. The value of this work lies in highlighting that generative systems also face this issue. Emphasizing this aspect would make the paper easier to follow and enhance its impact.

3. Lastly, while analyzing the smaller YFCC or CC datasets is commendable, I feel that modern VLMs are pretrained on a scale of billions. Therefore, presenting this analysis on these smaller datasets has limited practical impact compared to insights derived from larger datasets like LAION or DataCOMP. Additionally, YFCC and CC datasets have experienced data loss over time, further diminishing their relevance.

---

> ### Author Rebuttal · Authors · 2024-08-07
>
> > **W1: Dips in accuracy and consistency of log-linear trends?**
>
> We thank the reviewer for raising this concern.
> - **Analysis of drops in high freq concepts.** We provide some intuitions on why there are some drops in the trend at high freqs for CC-3M and CC-12M, we investigated which concepts occur at such high freqs, specifically above 10^4. The complete list of the relevant concepts and the corresponding downstream dataset will be added to the paper.
> From our analysis, we hypothesize two key reasons for a performance dip:
>   - **Concept ambiguity**: we observe many concepts that are homonyms / polysemous (same spelling but different meaning ie can represent multiple concepts at once). Some examples are watch, bear, house, fly, bridge, cloud, park, face, bar, tower, wave, etc.
>   - **Broad concepts**: A concept with a broader scope of definition supersedes a narrower one (concept ‘dog’ vs the specific breeds of dogs seen in imagenet-r ('yorkshire terrier', 'boston terrier', 'scottish terrier', 'golden retriever', etc)). These concepts are too coarse-grained and hence can be visually represented by a diverse set of images. Performance variance of these concepts can be quite high based on the specific set of images given for testing.
>
>   These ambiguities become more prevalent the more ubiquitous a concept is, which is directly tied to its freq obtained from pretraining datasets. Some more examples for a deeper understanding of the diversity of concepts are: cucumber', 'mushroom', 'Granny Smith', 'camera', 'chair', 'cup', 'laptop','hammer', 'jeep', 'lab coat', 'lipstick', 'american-flag', 'bear', 'cake', 'diamond-ring', etc.
>
> - **Statistical validity across broad evaluations.** Beyond our analysis, while we agree that there are some slight deviations from the log-linear scaling trend, particularly at high frequencies for the CC-3M and CC-12M datasets for classification, we would like to point out that these log-linear trends in general hold strongly across two different tasks and 4 different datasets. We have also validated that these trends are statistically significant by including the Pearson correlation coefficients and conducting a two-tailed t-test to increase confidence in our results.
>
> - **Noise in web-scale datasets and frequency estimation.** Further, we would like to highlight that web-scale datasets are inherently very noisy, and coupled with that our frequency estimation procedures could be quite noisy due to the inherent scale of our search and estimation procedures. Despite this noise, we observe strong log-linear scaling trends across the board. Further, we point the reviewer to the results in fig 22 of appx where we showcase that the log-linear scaling trends hold even when our frequency estimation pipeline is explicitly made noisier (by varying threshold of RAM++), further signifying the robustness.
>
> We will further ensure that each of the figure captions reflect the exact nature of the consistency of the log-linear trends in the main paper. We hope to have adequately addressed the reviewer’s concerns, by probing into the relevant concepts, as well as discussing the statistical validity of our results.
>
> > **W2: claiming first to show long-tailed VLM datasets, contributions with respect to prior work, and incorrect citation**
>
> We thank the reviewer for pointing out these concerns.
>
> - We apologise for the confusion—we would like to clarify that we **do not explicitly claim that we are the first to establish that VLM pretraining datasets are long-tailed**. We agree with the reviewer that prior work has showcased that VLM pretraining datasets like LAION-400M are long-tailed with respect to the concepts they contain, and we build, complement and generalize the findings of these prior works.
>
> - **Contributions with respect to [[1](https://arxiv.org/abs/2401.12425)]:** We emphasise that we complement this prior work and point out that our work comprehensively tests the strength of the log-linear scaling trend across several datasets, spanning varying levels of data curation and dataset sizes. Further, we note that in [1], the estimated frequencies are computed using only the text captions of LAION-2B. These estimated frequencies are then used as the canonical frequencies for plotting the performance-frequency curves for all the tested models (despite these models being trained on different pretraining datasets other than LAION-2B). Our work strongly showcases why this apparent asymmetry in their frequency estimation methodology should work—from tab 4, we show that different VLM pretraining datasets are strongly correlated in their concept distributions. Hence, in spite of [1] using only LAION-2B as their source dataset for freq estimation, their results roughly hold true because of this strong correlation across pretraining datasets. Our methodology of incorporating both images and text captions when computing the freq estimates is crucial for explaining this. Hence, we believe that our work comprehensively generalizes and explains the findings of prior work while also providing insights into the pretraining datasets (eg, misalignment degree and correlation of concept distributions in datasets). We will be sure to clarify this in the revision.
>
> - **Scale of experiments:** While we agree that the largest dataset we analyse (LAION-400M) is smaller than LAION-2B/DataComp-1B, we point to fig 3 and 22 of [[2](https://arxiv.org/abs/2304.14108)], where authors showcase that zero-shot performance of different data curation methods are strongly correlated across different dataset scales (12.8M vs 128M vs 1.28M), while the source distribution is held constant (Common-Crawl). Similarly, in our case since LAION-400M, LAION-2B and DataComp-1B are sourced from the same Common-Crawl source-distribution, we are confident our results also generalize to LAION-2B / DataComp-1B.
>
> - **Incorrect citation:** We apologise for this oversight—we fixed it in the paper to reflect the correct version.

---

> ### Author Response · Authors · 2024-08-07
> **Rebuttal to reviewer WCHb contd. [1/2]**
>
> > **W3: info on Let-It-Wag! construction**
>
> Thank you for pointing out this concern, we apologise for the lack of details regarding the construction of the Let-It-Wag! Benchmark. We provide some details here and have also added this information to the appx.
>
> For curating the set of 290 concepts, we collate our frequency estimates from the LAION-400M dataset for all the 4,029 concepts we consider. We then remove all the concepts that have 0 counts to ensure that our final dataset consists of concepts that have been detected atleast once in LAION-400M, this method has also been used in [[3](https://arxiv.org/pdf/2211.08411)] to ensure robustness to noise in the estimation process. For each extracted concept, we then apply the image download and manual cleaning procedure described in appx G.2. Finally, we removed those concepts that had a low number of total images left post the cleaning process, and retained all the other concepts. We then sorted these concepts in ascending order of their estimated concept frequencies, and retained the top 290. These concepts are used as the final list of classes for our Let-It-Wag! benchmark.
>
> > **W4: high-level insights about long-tail concepts**
>
> Thank you for this question. The broad categories of the most long-tailed concepts with a few examples for each are as follows (a majority of them have been highlighted already in Figs 24-27 in the appendix):
> - **Birds:** Western_Scrub_Jay, Cassins_Finch, Prairie_Warbler, Red_eyed_Vireo, Veery
> - **Animals:** flatworm, Tibetan_Mastiff, Scottish_Terrier, vine_snake, newt
> - **Aircrafts:** A300B4, A310, Falcon_900, DHC-8-300, MD-11
> - **Objects:** guillotine, letter_opener, ladle, dust_jacket
> - **Plants and fungi:** mexican_aster, gyromitra, great_masterwort, thorn_apple, cape_flower
> - **Misc:** consomme, stratified_texture, eggnog
>
> Additionally, we will release the full list of these concepts as the classes of the Let-It-Wag! benchmark in the appx.
>
> > **Q1: stats about Let-It-Wag?**
>
> Thank you for raising this question. We provide the required stats below.
> - **Most frequent concepts:** partridge (count=9489), Bank Swallow (count=9489), eel (7907)
> - **Least frequent concepts:** Red-necked Grebe (count=0), SR-20 aircraft (count=0), Globe-flower (count=0)
> - **Median frequency of concepts:** 97.5
> - **Mean frequency of concepts:** 1096.2
>
> We also show the full histogram of concept frequencies for the 290 concepts in the Let-It-Wag! dataset in the uploaded rebuttal pdf. From the histogram, it is evident that most of the concepts in Let-It-Wag! have frequency less than 2000. About half of the concepts in Let-It-Wag! (~140) have a frequency less than 1000. Hence, this histogram sufficiently establishes that our **Let-It-Wag! dataset truly captures the long tail**.
>
> > **Q2: variance in fig 2 points**
>
> Thank you for the insightful question. We provide zero-shot classification plots for CC-3M, CC-12M, and LAION-400M in the uploaded rebuttal PDF, including 95% confidence intervals for each point. This approach follows the standard practice from works like [[1](https://arxiv.org/abs/2107.04649), [2](https://arxiv.org/abs/2007.00644)]. Our plots show that **the spread at higher frequencies is significantly larger than at moderate frequencies**, corroborating the finding that **higher frequency concepts are more ambiguous and polysemous**. These results **support the observed dips in accuracy at high-frequency points**. We will include these plots in the appendix and discuss this in the paper.

---

> ### Author Response · Authors · 2024-08-07
> **Rebuttal to reviewer WCHb contd. [2/2]**
>
> > **L1: mitigating solutions?**
>
> Thank you for this insightful question. While our paper does not propose specific solutions, we believe its primary contribution is in thoroughly highlighting the issues with current pretraining strategies for multimodal models across various datasets, pretraining methods, architectures, training objectives, and tasks. Additionally, by releasing the "Let it Wag!" testbed, we provide a straightforward test set for future research to build upon, aiming to improve the generalization of multimodal models to long-tail scenarios.
> However, we suggest a few potential methods that could be explored to enhance multimodal long-tail:
>
> - **Retrieval Augmentation:** Enhancing generalization to long-tail concepts can be achieved by utilizing the "world-knowledge" of LLMs to provide detailed descriptions for these concepts. This approach transforms the task from simply recognizing long-tail concepts by name to recognizing them by both names and descriptions.
> - **Curriculum Learning:** Our tested models used random IID sampling during training. However, research into better sequencing of data samples could potentially improve model generalization to long-tail concepts by inducing more transferable feature representations in VLMs.
> - **Synthetic Data:** Addressing the issue of long-tail concepts in web-sourced datasets may not be feasible by merely increasing data samples. There will likely always be low-data density regions in the pretraining data distribution. Using synthetic data, either through procedurally generated samples or text-to-image models, could be a viable mitigation strategy.
>
> We hope these suggestions provide valuable directions for future research and contribute to the development of multimodal models capable of better generalization. We will add these points to the future works section in the appendix.
>
> > **L2: stronger focus on generation and Let-It-Wag! dataset**
>
> We thank the reviewer for raising this point. We would like to respectfully argue that we believe our image-text contributions are equally as important to the other contributions, please see response to W2. Re. focus on Let-It-Wag! dataset, we agree that it would be prudent to emphasize its construction/properties further. We will make these points clearer in the paper and hope this clarifies the issue.
>
> > **L3: usefulness of insights gained from CC/YFCC experiments?**
>
> We thank the reviewer for raising this important point.
>
> - **CC/YFCC datasets used for important VLM pretraining methods.** While we agree that smaller datasets like CC-3M/YFCC-15M might not yield insights which are practically relevant for SoTA performance, we still believe that it is an important validation to perform these experiments at that scale. We note that lots of high impact work on CLIP-like models have been empirically validated by pretraining on datasets like CC-3M/CC-12M/YFCC-15M, including [CyCLIP](https://arxiv.org/abs/2205.14459), [DeCLIP](https://arxiv.org/abs/2110.05208), [OpenCLIP](https://github.com/mlfoundations/open_clip), [SLIP](https://arxiv.org/abs/2112.12750), [FILIP](https://arxiv.org/abs/2111.07783), [MERU](https://arxiv.org/abs/2304.09172) and [LaCLIP](https://arxiv.org/abs/2305.20088).
>
> - **Expansive insights spanning a range of dataset scales.** Further, we note that each of these datasets from small-scale CC-3M to the large scale LAION-400M employ different data curation strategies, and by showcasing that (1) *all datasets have very similar pretraining concept distributions*, and (2) *models trained on any of these datasets showcase the same consistent log-linear scaling trends*, we have empirically showcased the robustness of our main exponential data inefficiency result, and uncovered several important properties of VLM pretraining datasets.
>
> - **Significant compute resources.** Finally, we would like to point out the significant compute resources it would take to conduct our analysis on even larger-scale datasets like DataComp-1B or LAION-2B—in appx L tab 8, we report the storage and compute costs required for conducting all our experiments—at the LAION-400M scale, we would need almost 10TB of disk space and 2000GPU / 7000CPU hours (this is roughly in the order-of-magnitude required for pretraining CLIP models themselves, see [Cherti et al](https://arxiv.org/abs/2212.07143) tab 18).
>
> - **Testing on other web-scale datasets with Let-It-Wag!** Given our results’ robustness across several scales, we have strong reasons to believe that our result will continue to generalize even for larger-scale datasets and models. One point of evidence to further bolster this is that we tested models trained on larger-scale datasets including DataComp-1B, DFN-1B and WebLI-10B, in our analysis on the Let-It-Wag! Dataset in fig 6. We however do not see any significant deviations for models trained on these datasets that would lead us to believe that they showcase different characteristics to the datasets we analysed in our work.

---

> ### Comment · Reviewer_WCHb · 2024-08-11
>
> Thank you for your response—it has addressed most of my questions. Here are my thoughts on each point:
>
> 1. I think it's important to note the ambiguity problem as a limitation of the current frequency measurement pipeline. This can help offer some explanation of the dips observed and clear some questions a reader may have when observing the dips.
>
> 2. It seems there's still some confusion regarding [1], which also analyzes LAION-400M, not just LAION-2B, as I mentioned in my review. Since your work is closely related to previous research, it's crucial to provide a clear comparison that differentiates your contributions from existing work. I suggest enhancing the current literature review section to better highlight these novel aspects, as the current writing doesn't achieve this.
>
> 3. Thank you for detailing the dataset curation process. **However, I'm curious if you considered reducing the benchmark size by sampling only classes with fewer than 1,000 instances. If so, did this result in further accuracy degradation?**
>
> 4. It would be beneficial to include the dataset statistics and high-level insights presented in the rebuttal within the appendix of the submission.
>
> I have asked one additional question above and will wait for the authors' reply.

---

> > ### Author Response · Authors · 2024-08-12
> > **Response to Reviewer WCHb**
> >
> > Thank you for following up, and highlighting what should be resolved in the revision—we will most certainly fix the references, will point to our ambiguity analysis in the main text, ensure our contributions wrt to [1] are adequately addressed in the literature review, and altogether will include all rebuttal insights within the appendix of the revised paper.
> >
> > Regarding creating a filtered version of the Let-It-Wag! dataset, as suggested, we keep only those classes that have a frequency of less than or equal to 1000 instances per class. This filtered dataset contains 151 classes compared to the original 290 classes in Let-It-Wag!. We then re-ran a diverse set of zero-shot CLIP/SigLIP models on this Let-It-Wag-filtered dataset, and showcase the comparison results to both ImageNet and the original Let-It-Wag! dataset in the table below:
> >
> > |Model/Dataset|ImageNet Acc|Let-It-Wag! Acc (290 classes)|Let-It-Wag-filtered! Acc (151 classes)|
> > |-|-|-|-|
> > |RN50/CC-3M|20.09|3.74|2.56|
> > |RN50/CC-12M|33.14|8.92|5.95|
> > |RN-50/openai|59.84|32.93|30.18|
> > |ViT-B-32/openai|63.34|33.52|32.90|
> > |ViT-B-32/datacomp|69.18|46.91|49.21|
> > |ViT-B-16/CC-3M|17.11|3.01|2.42|
> > |ViT-B-16/CC-12M|37.39|11.49|7.88|
> > |ViT-B-16/openai|68.36|37.86|37.77|
> > |ViT-B-16/datacomp|73.48|52.90|56.00|
> > |ViT-B-16/WebLI|78.49|54.64|49.69|
> > |ViT-L-14/openai|75.53|45.32|44.49|
> > |ViT-L-14/datacomp|79.21|63.04|65.70|
> > |ViT-H-14/DFN|83.45|71.91|71.80|
> > |ViT-L-16/WebLI|82.07|61.51|56.13|
> > |ViT-So400m/WebLI|82.03|67.33|63.61|
> >
> > We observe that almost all models underperform on the Let-It-Wag-filtered dataset compared to the Let-It-Wag! dataset. One interesting point to note here is that all exceptions were models trained on DataComp-1B---they perform ~2-3% worse on Let-It-Wag! compared to the Let-It-Wag! filtered dataset. Given DataComp’s focus of optimizing data curation, this perhaps suggests that research into better data curation could be a viable route to enabling progress on the long-tailed concepts, however this warrants a more concrete treatment which could be an interesting direction for follow-up work to explore. With that said, **all performance numbers remain significantly lower than ImageNet performance across all models, which underscores that models still struggle to perform well on concepts that are long-tailed in web data-distributions**. We will highlight these results and interpretations in the revision.
> >
> > We hope this sufficiently addresses the reviewer's concerns and are happy to provide further clarifications if required.

---

> > > ### Comment · Reviewer_WCHb · 2024-08-12
> > >
> > > I noticed that the above results don't include those for LAION-400M. Since the analysis is based on LAION-400M, including these results for a more comprehensive overview might be helpful.

---

> > > > ### Author Response · Authors · 2024-08-12
> > > >
> > > > Thanks for highlighting this, we've included the results for models pretrained on LAION-400M in the table below, we hope this sufficiently addresses the reviewer's concerns.
> > > >
> > > > |Model/Dataset|ImageNet Acc|Let-It-Wag! Acc (290 classes)|Let-It-Wag-filtered! Acc (151 classes)|
> > > > |-|-|-|-|
> > > > |ViT-B-32/LAION-400M|60.24|34.02|32.88|
> > > > |ViT-B-16/LAION-400M|67.03|40.53|39.14|
> > > > |ViT-L-14/LAION-400M|72.75|48.10|46.59|

---

> > > > > ### Comment · Reviewer_WCHb · 2024-08-12
> > > > >
> > > > > Thank you for addressing my concerns; I am satisfied with the authors' responses.
> > > > >
> > > > > If the authors can incorporate the feedback from this rebuttal, such as updating the related work, discussing the dips in high-frequency classes, and including details for the 'Let It Wag' dataset, the paper would be a strong addition to NeurIPS 2024. Foundational models like CLIP have gained significant popularity due to their zero-shot recognition capabilities, so identifying their limitations is crucial.
> > > > >
> > > > > In light of this discussion and the comprehensive answers provided, I am increasing my score from 6 to 7. I appreciate the authors' efforts and hope they will incorporate the suggested changes.

---

> > > > > > ### Author Response · Authors · 2024-08-12
> > > > > >
> > > > > > We would like to thank the reviewer for deeply engaging with our work and helping us improve the overall quality of the paper. We will most definitely take into account all the points raised by the reviewer (regarding related work, discussing high-frequency dips, and statistics of Let-It-Wag!) and update the pertinent sections in the revised version of the paper.

---

### Official Review · Reviewer_s2XF · 2024-07-13

**Soundness:** 3
**Presentation:** 4
**Contribution:** 4
**Rating:** 8
**Confidence:** 4

**Summary:**

This paper examines the relationship between the frequency of concepts in pretraining data and the performance of downstream tasks associated with those concepts. Extensive experimental results reveal a log-linear relationship, suggesting that exponential increases in data are necessary to improve zero-shot model performance. Additionally, the authors reconfirm the long-tailed distribution of concepts in well-known pretraining datasets, highlighting the challenge of handling rare concepts in foundation models. Based on this analysis, the authors provide a benchmark to evaluate model performance on tail concepts.

**Strengths:**

- This paper significantly advances our understanding of data efficiency in multimodal foundation models, revealing a log-linear relationship between concept frequency in pretraining datasets and downstream task performance.
- Extensive experiments are conducted, covering various pretraining datasets and downstream tasks.
- The additional analysis provides valuable insights into the long-tailed distribution problem of foundation models.
- The paper introduces a new benchmark called “Let It Wag,” crucial for evaluating the performance of multimodal foundation models on long-tail distributions.
- The paper is clearly written and easy to follow, with most relevant details included.

**Weaknesses:**

Nothing I can think of.

**Questions:**

- I am curious about the phenomenon where the average accuracy drops after the concept frequency of 10^4 in Figure 2. Could the authors provide some examples and hypotheses about why this happens?
- How much variance is there in each point of Figure 2? I am curious about how much the tendency varies depending on the difficulty of the concept, its size, or other aspects.
- Does zero-shot performance predict few-shot or linear probing performance? I wonder if zero-shot models still learn important features for rare concepts, but these features are dominated by class imbalance in zero-shot tasks.
- Typically, how long does it take to train a model on each dataset? (GPU hours)

**Limitations:**

The authors adequately addressed the limitations with future directions in Appendix N.

---

> ### Author Rebuttal · Authors · 2024-08-07
>
> > **Q1: Dips in accuracy at high freqs**
>
> Thank you for the suggestion. We look into CC-3M and CC-12M, the pretraining datasets corresponding to which we see dips in accuracy on the classification tasks. From our analysis, we hypothesise two main reasons for these performance dips:
>
> - **Concept ambiguity:** we observe many concepts that are homonyms / polysemous (same spelling but different meaning ie can represent multiple concepts at once). Some examples are watch, bear, house, fly, bridge, cloud, park, face, bar, tower, wave, etc.
> - **Broad concepts:** A concept with a broader scope of definition supersedes a narrower one (concept 'dog' vs the specific breeds of dogs seen in imagenet-r ('yorkshire terrier', 'boston terrier', 'scottish terrier', 'golden retriever', etc)). These concepts are too coarse-grained and hence can be visually represented by a diverse set of images. Performance variance of these concepts can be quite high based on the specific set of images given for testing.
>
> These ambiguities become more prevalent the more ubiquitous a concept is, which is directly tied to its frequency obtained from pretraining datasets. Some more examples for a deeper understanding of the diversity of concepts are: cucumber', 'mushroom', 'Granny Smith', 'camera', 'chair', 'cup', 'laptop','hammer', 'jeep', 'lab coat', 'lipstick', 'american-flag', 'bear', 'cake', 'diamond-ring', etc. We will provide this analysis in the appendix and hope this adequately addresses the question.
>
> > **Q2: Variance in fig 2 points**
>
> Thank you for the insightful question.
> We provide zero-shot classification plots for CC-3M, CC-12M, and LAION-400M in the uploaded rebuttal PDF, including 95% confidence intervals for each point. This approach follows the standard practice from works like [[1](https://arxiv.org/pdf/2107.04649), [2](https://arxiv.org/abs/2007.00644)]. Our plots show that **the spread at higher frequencies is significantly larger than at moderate frequencies**, following the finding in the previous point that **higher frequency concepts are more ambiguous and polysemous**. These **results support the observed dips in accuracy at high-frequency points**. We will include these plots in the appendix and discuss this in the paper.
>
> > **Q3: Zero-shot perf predictive of few-shot/linear-probing?**
>
> Thank you for this insightful question. Our work focused on zero-shot evaluations, the standard for assessing vision-language models "out-of-the-box". However, according to [Gadre et al.](https://arxiv.org/abs/2304.14108) fig. 16, ImageNet zero-shot performance and linear probing performance are highly correlated across various CLIP models. Thus, it is likely our trends would also apply to few-shot fine-tuning, at least for ImageNet.
>
> We agree with the reviewer that investigating how log-linear scaling trends change with few-shot fine-tuning, such as in [TIP-Adapter](https://arxiv.org/abs/2111.03930), [CLIP-Adapter](https://arxiv.org/abs/2110.04544), or [CoOP](https://arxiv.org/pdf/2109.01134), would be a valuable follow-up to our work. We will add a point on this in our future works section in the appendix.
>
> > **Q4: Training time for models?**
>
> Each of the models we consider are roughly trained for ~30 epochs each. This results in a different total compute budget (samples seen budget) for each run. We provide some canonical estimated total GPU hours for training different models below:
>
> |Model|Samples seen|GPU hours|
> |-|-|-|
> |RN50|90M/340M/430M|50h/186h/240h|
> |ViT-B-32|13B|4500h|
> |ViT-B-16|90M/340M/430M/13B|60h/200h/280h/5000h|
> |ViT-L-14|13B|7000h|

---

> ### Comment · Reviewer_s2XF · 2024-08-09
>
> Thank you for your further investigation and thorough response! My questions are nicely addressed, and I really enjoyed reading your paper. I am going to keep my score as it is, since I already gave a high score from the beginning.

---

> > ### Author Response · Authors · 2024-08-12
> >
> > We would like to thank the reviewer for deeply engaging with our work and helping us improve the overall quality of the paper.

---

### Official Review · Reviewer_oRi7 · 2024-07-15

**Soundness:** 3
**Presentation:** 3
**Contribution:** 2
**Rating:** 6
**Confidence:** 4

**Summary:**

This work explores the extent to which zero-shot generalisation really occurs in large-scale in model that were trained on web-scale datasets. The approach taken relies on identifying concepts that are present in train and test data and evaluating concept frequencies and per-concept performance. From extensive experiments, the authors find that test performance of the models correlates strongly with the frequency of concepts seen during training.

**Strengths:**

* The paper tackles an interesting and important topic - something that is often mentioned / discussed, but prior to this paper did not get proper analytic treatment.
* Good and mostly clear presentation, clear methodology, extensive experimental evidence

**Weaknesses:**

* My general feel after reading the paper is that the issue of zero-shot performance of model trained on web-scale data is not as bad as the paper (abstract) makes it out to be. For example, from Fig 6 we see that ImageNet and long-tail (e.g. Let It Wag!) benchmarks actually tend to agree as model performance improves. Similarly, we see from Fig 2 that larger models tend to have better performance even for concepts with low train dataset frequencies.
* Presentation of the results is not always clear. For example, what are the different pannels in Fig 3 or Fig 5?
* Results presented in the paper do not really study zero-shot performance. Specifically, in Fig 2 or 3 there appears to be no evaluated points with freq = 0. As such, the paper really studies few-shot performance as a function of the number of training examples.
* Methodologically, I have an issue with way concept frequency in images was estimated - it relies on a pre-trained model (RAM++) to tag/classify images. This model itself likely suffers from training datasets biases and thus could miss or over- or under-estimate image concepts. This limitation also puts into questions the image-caption misalignment results presented in Tab 3.

**Questions:**

See weaknesses.

**Limitations:**

See weaknesses, specifically whether zero-shot performance is really being evaluated in the paper, and how concept frequencies were determined for images.

---

> ### Author Rebuttal · Authors · 2024-08-07
>
> > **W1: Why not simply scale up models?**
>
> Thank you for raising this point.
> - **Train-test similarity as a control factor.** We agree models with higher ImageNet accuracy perform better on the long-tail. Note however that results in fig 6 are not normalized for *“train-test similarity”*. Normalizing for *train-test similarity* is crucial for interpreting these results. For eg, the absolute performance on low-freq concepts is higher across larger datasets like LAION400M in fig 2. However, when normalized for train-test similarity (fig 4 left), performance drops to levels comparable to smaller datasets like CC-12M. This highlights the importance of considering *train-test similarities* when comparing absolute performance across pretraining datasets.
> - **On scaling up models.** We agree increasing model sizes can improve performance offsets, as supported by scaling laws. However, trends in performance degradation on low-freq concepts remain consistent across different model-sizes (RN50 - ViT-L/14), indicating a significant issue even with larger models.
>
> We will clarify these points in the manuscript and add a discussion on model scaling laws.
>
> > **W2: What are panels in Fig 3/Fig 5?**
>
> We apologize for the lack of clarity.
> - **Figure 3**: The different panels denote log-linear performance-frequency scaling curves for different T2I models. Note that in total we analyze 24 different T2I models (see tab 2). Since it would be tedious to fit all 24 onto a single plot, we split into 8 sub-plots with 3 models each.
> - **Figure 5**: The different panels in fig 5 all showcase the concept count distribution across three pretraining datasets. In each plot, we showcase estimated frequency of concepts in the pretraining dataset on the y-axis, and the sorted concept index (sorted by frequency) on the x-axis. Since there are three ways to estimate concept frequencies (freqs from text captions, freqs from images, and joint frequencies from both images and text captions), we showcase the frequency distributions as obtained using all three methods independently.
>
> We will update the paper to add these clarifications, and hope this will simplify and ease the presentation of results.
>
> > **W3: Paper does not really study 0-shot performance, no points with freq 0 in figs 2 and 3**
>
> Thank you for this important point. The reviewer is right—we explicitly exclude zero-freq concepts from our evaluations following [[1]](https://arxiv.org/pdf/2211.08411), since frequency estimation is potentially noisy, leading to low recall rates (also discussed in appx H). However, to verify if our log-linear trends still hold when including all the zero-freq concepts, we replot all our main zero-shot classification results from fig 2 by including the ones which have zero-freqs—this plot is in the attached PDF. **We find our main log-linear scaling trends from fig 2 are retained**.
> To further corroborate this, we present average accuracies for concepts with freq 0 and non-zero freq bins in the table below. We note that **average performance for the 0-freq concepts are significantly lower than other non-zero freq concepts**, especially when compared to very high-freq concepts. This justifies our main claim that exponentially more data is needed per concept to improve performance linearly.
> |Dataset/Model|freq=0|freq=1-10|freq=10-100|freq=100-1000|freq=1000-10000|
> |-|-|-|-|-|-|
> |CC-3M/RN50|5.10|13.89|20.18|32.93|44.30|
> |CC-3M/ViT-B-16|4.27|11.98|17.21|27.48|39.24|
> |CC-12M/RN50|12.91|21.49|27.75|39.48|50.38|
> |CC-12M/ViT-B-16|16.48|25.59|32.07|45.65|57.06|
> |YFCC-15M/RN50|16.49|19.59|24.12|34.26|39.97|
> |YFCC-15M/RN101|17.43|22.06|25.72|36.77|43.14|
> |YFCC-15M/ViT-B-16|20.75|25.06|29.68|38.73|45.96|
> |LAION-400M/ViT-B-32|47.41|46.42|50.53|55.96|65.00|
> |LAION-400M/ViT-B-16|51.77|52.09|57.12|61.32|70.73|
> |LAION-400M/ViT-L/14|60.44|58.87|62.43|67.63|76.65 |
>
> > **W4: Issues with RAM++**
>
> Thank you for this important point.
> - **Extensive Ablations on Image-Tagging Models:** We agree using a pretrained model for tagging concepts might introduce biases. However, we conducted extensive ablations on this (see appx H). We tested concept-tagging ability of open-world object detectors (Owlv2) and multi-label tagging models (RAM/RAM++), finding RAM++ to be most precise for our case (see appx H.1, fig 20).
> - **Context Enhancement improves RAM++ tagging precision**: Unlike object detectors, RAM++ leverages GPT-4-generated descriptions (see tab 5 appx), improving tagging precision by using visual descriptions to better identify concepts (this has been shown to enhance performance [[2](https://arxiv.org/abs/2210.07183),[3](https://arxiv.org/abs/2209.03320)].
> - **Robustness vs RAM++ Thresholds:** We investigated different hparam thresholds for RAM++ (appx H.2). Despite some thresholds yielding sub-optimal tagging, our log-linear scaling results remained robust.
> - **Human verification for misalignment results**: To verify misalignment results, we manually annotated 200 random image-text pairs from each dataset as aligned or misaligned. An image-text pair is misaligned if the text caption was irrelevant to the image. Previous work also found a similarly small random subset over large-scale web-datasets to be representative [[4]](https://arxiv.org/abs/2307.03132). Our estimated misalignment results from tab 3 were in line with human-verified results (see tab below), corroborating our findings.
> |Dataset|tab 3 results|human baseline results|
> |-|-|-|
> |CC-3M|16.81%|18%|
> |CC-12M|17.25%|14.5%|
> |YFCC-15M|36.48%|40.5%|
> |LAION-400M|5.31%|7%|
> - **High YFCC Misalignment Degree:** From our human experiment, we found that the high misalignment degree in YFCC-15M is likely due to the lack of text quality filtering. YFCC-15M images are sourced directly from Flickr, where captions often provide high-level context rather than accurately describing the image content.
>
> We hope these clarifications address the reviewer's concerns and provide a better understanding of our work.

---

### Author Rebuttal · Authors · 2024-08-07

**General Response to all reviewers**

We thank all the reviewers for finding our work ***interesting and important*** (Reviewer oRi7), ***clearly written and well presented*** (Reviewers oRi7, s2XF, WCHb, KUaM), ***containing extensive empirical evidence*** (Reviewers oRi7, s2XF, KUaM), and for finding ***our Let-It-Wag benchmark useful*** (Reviewers s2XF, WCHb). We provide detailed answers to each of the individual reviewers' concerns independently, and collate the most important common points here to further reiterate additional experimental results provided during the rebuttal.

1. We have added plots that include the **0-frequency concepts in the zero-shot classification plots** in the uploaded rebuttal pdf. We find that even when incorporating the significantly noisier 0-frequency concepts into our plots, our **main log-linear performance-frequency scaling trends remain preserved**.

2. We have provided intuitions for why there is an apparent dip in performance at the high frequency concepts for CC-3M and CC-12M. We hypothesize that these high-frequency concepts are **homonyms/polysemous** and **broad**, suggesting that the concept difficulty and the visual diversity of related test-set images of these high-frequency concepts is much more varied.

3. We have added **variance plots for showcasing the spread across each point in the zero-shot classification results** by including 95% confidence intervals. Our plots show that the **spread at higher frequencies is significantly larger than at moderate frequencies**, corroborating the finding that **higher frequency concepts are more ambiguous and polysemous**. This further explains some of the dips in performance we see at higher frequency concepts.

4. We have additionally provided more detailed statistics on the Let-It-Wag! dataset. We provide a histogram that showcases the exact tailed nature of our dataset—the median frequency of concepts in the dataset is less than 100, suggesting that **our Let-It-Wag! dataset truly tests the long-tail**.

5. We have run additional experiments with both SLIP and CyCLIP models, both of which claim to improve the generalization of CLIP models. We find that even **for these models with different training objectives, the log-linear scaling trends still hold true**, suggesting that they do not fully close the gap to improving the long-tailed performance.

---

### Decision · Program_Chairs · 2024-09-25

**Decision:**

Accept (poster)

**Comment:**

This work conducted a systematic study on the zero-shot generalization capability of multimodal models such as CLIP for classification and Stable Diffusion for image generation. It receives unanimous positive scores after rebuttal discussions. The problem of understanding the limitations of VLMs is important, the study on the T2I models is insightful, and the dataset that aggregates the tailed concepts is valuable. AC agrees this is a solid contribution to the community and recommends acceptance. The authors should properly acknowledge and discuss prior work "The Neglected Tails in Vision-Language Models" as suggested by the reviewers.